# The DNA demethylase TET1 modifies the impact of maternal folic acid status on embryonic brain development

Lehua Chen [1,5], Bernard K van der Veer [1,5], Qiuying Chen [2], Spyridon Champeris Tsaniras[1], Wannes Brangers[1], Harm H M Kwak[1], Rita Khoueiry [1], Yunping Lei[3], Robert Cabrera[3], Steven S Gross [2], Richard H Finnell [3,4] & Kian Peng Koh [1,3]✉

## Abstract

Folic acid (FA) is well known to prevent neural tube defects (NTDs), but we do not know why many human NTD cases still remain refractory to FA supplementation. Here, we investigate how the DNA demethylase TET1 interacts with maternal FA status to regulate mouse embryonic brain development. We determined that cranial NTDs display higher penetrance in non-inbred than in inbred *Tet1*[−/−] embryos and are resistant to FA supplementation across strains. Maternal diets that are either too rich or deficient in FA are linked to an increased incidence of cranial deformities in wild type and *Tet1*[+/−] offspring and to altered DNA hypermethylation in *Tet1*[−/−] embryos, primarily at neurodevelopmental loci. Excess FA in *Tet1*[−/−] embryos results in phospholipid metabolite loss and reduced expression of multiple membrane solute carriers, including a FA transporter gene that exhibits increased promoter DNA methylation and thereby mimics FA deficiency. Moreover, FA deficiency reveals that *Tet1* haploinsufficiency can contribute to DNA hypermethylation and susceptibility to NTDs. Overall, our study suggests that epigenetic dysregulation may underlie NTD development despite FA supplementation.

**Keywords** Folic Acid; DNA Methylation; Gene-environment; Embryonic Brain; Neural Tube Closure
**Subject Categories** Chromatin, Transcription & Genomics; Development; Metabolism

## Introduction

Neural tube defects (NTDs) are among the most common congenital malformations, affecting 0.3–200 per 10,000 pregnancies globally with regional variation in prevalence (Zaganjor et al, 2016). These birth defects are caused by a failure of the neurulation process early in embryogenesis, when the neural plate folds into a tube to form the morphological scaffold for the developing central nervous system (Nikolopoulou et al, 2017; Wallingford et al, 2013; Yamaguchi and Miura, 2013). Failure of neural tube closure (NTC) at the cranial and spinal regions results in the most prevalent open NTD subtypes—anencephaly and spina bifida, respectively (Copp and Greene, 2010). The etiology of NTDs is currently postulated to be multifactorial with contributions from both genetic and environmental factors (Finnell et al, 2021; Juriloff and Harris, 2018; Wilde et al, 2014). Maternal folate (vitamin B9) status is the major nutritional factor known to modify NTD risks (Group, 1991). Supplementation of folic acid (FA), the synthetic form of folate, at 0.4 to 0.8 mg daily periconceptionally has a substantial net benefit to prevent NTDs, but 30–50% of cases remain non-responsive to FA (Crider et al, 2022). Despite aggressive campaigns for FA supplementation and fortification of the US food supply, NTDs still affect an estimated 3000 pregnancies in the US annually and remain a significant global public health concern (Force et al, 2023). Recent studies in rodent models indicate that excessive FA supplementation during pregnancy can also result in neuronal deficits in the developing brain (Harlan De Crescenzo et al, 2021; Wang et al, 2021). Therefore, it is important to understand the consequences of both surplus and deficient maternal FA intake on offspring health.

In mouse models, NTDs are most frequently observed as exencephaly, the developmental precursor of anencephaly (Copp and Greene, 2010; Leduc et al, 2017). Over 300 genes, when mutated in the mouse, are known to result in NTDs (Harris and Juriloff, 2010). However, NTD phenotypes in single gene knockout (KO) mouse models are often incompletely and lowly penetrant, suggesting a polygenic etiology in the disease (Leduc et al, 2017; Rai et al, 2023). NTDs in a few mouse KO models have been reported to alter penetrance in different strain backgrounds, indicating an important contribution of genetic modifiers in their phenotypic

[1]Department of Development and Regeneration, Stem Cell and Developmental Biology, KU Leuven, Leuven 3000, Belgium. [2]Department of Pharmacology, Weill Cornell Medical College, New York, NY 10065, USA. [3]Department of Molecular and Cellular Biology, Center for Precision Environmental Health, Baylor College of Medicine, Houston, TX, USA. [4]Department of Molecular and Human Genetics, Department of Medicine, Baylor College of Medicine, Houston, TX, USA. [5]These authors contributed equally: Lehua Chen, Bernard K van der Veer. ✉E-mail: kian.koh@kuleuven.be

expression. For example, *Sall2* and *Men1* null mouse models showed 10% and 20% penetrance of NTD in 129 strains, respectively, but the phenotype was completely absent in the C57BL/6 strain (Harris and Juriloff, 2010). While the mouse remains the most common animal model to study the role of candidate genes in the development of NTDs, the predominant use of the C57BL/6J (abbreviated as B6) congenic background may not adequately model the complex genetic landscape of NTDs. Thus, a comprehensive understanding of any genetic and non-genetic factor in NTDs using mouse models may require phenotypic characterization in alternative strains.

As an essential nutrient, folate is a key co-factor in a network of interlinked metabolic reactions that provide one-carbon (1C)-units for nucleotide biosynthesis and methylation reactions (Crider et al, 2012). In the latter, the folate cycle feeds into the methionine cycle to generate S-adenosylmethionine (SAM), a universal methyl-donor required for methylation of DNA, RNA, and proteins. Human and animal studies suggest that folate status influence global patterns of DNA methylation, which occurs predominantly at CpG dinucleotides in mammalian DNA (Cao et al, 2023; Irwin et al, 2016; Joubert et al, 2016). Maternal folate deficiency may increase the risk of NTD by inducing DNA hypomethylation (Chang et al, 2011; Chang et al, 2021; Lu et al, 2022; Wang et al, 2017). Mice deficient in the de novo DNA methyltransferase 3A (DNMT3A) and DNMT3B, showed an increased risk for NTDs and other embryonic malformations, indicating that impaired DNA methylation can disrupt NTC (Okano et al, 1999).

The role of active DNA demethylation during early embryogenesis has been clarified more recently with the discovery of the Ten-Eleven-Translocation (TET1, TET2, and TET3) DNA dioxygenases. TET enzymes catalyze DNA demethylation via the iterative oxidation of 5-methylcytosine (5mC) into 5-hydroxymethylcytosine (5hmC) and further oxidation products, which are subsequently replaced with unmethylated cytosines, either passively by DNA replication or actively by base excision repair (He et al, 2011; Ito et al, 2011; Tahiliani et al, 2009). We have previously demonstrated a dominant role of TET1 in the mouse embryo early post-implantation, involving both catalytic and non-catalytic activities in lineage gene regulation (Khoueiry et al, 2017). While high *Tet1* expression is sustained in the epiblast until the onset of gastrulation, loss of TET1 results in both repressive DNA and histone hypermethylation that persist in post-gastrulation neuroepithelial cells, ultimately resulting in cranial NTDs (Luo et al, 2020; van der Veer et al, 2023). Intriguingly, the penetrance of embryonic defects upon *Tet1* ablation originally observed in a gene trap (GT) strain is strongly influenced by genetic backgrounds, being partial in a B6 congenic strain but completely lethal in outbred stocks (Khoueiry et al, 2017). This prompted us to investigate whether TET1 may be an epigenetic regulator at the nexus of gene-environmental interactions linked to folate 1C metabolism.

In this study, we characterized the phenotypes of *Tet1* null embryos in a gene targeted mouse model across genetic backgrounds, including the congenic B6, outbred CD1 and an alternative 129S6 strain, and the impact of modifying maternal dietary FA status. Post-NTC mouse embryos collected at E11.5 were further analyzed by high-throughput profiling of the whole-embryo metabolome and embryonic brain DNA methylome and transcriptome. We determined *Tet1* gene dosage interactions with folate status contributing to NTD susceptibility, resistance to FA supplementation, and differential DNA methylation patterns that converge on the regulation of folate uptake, phospholipid metabolism, and neurotransmitter functions.

## Results

### Loss of *Tet1* results in NTDs with higher penetrance in non-inbred than in inbred genetic backgrounds

In our previous work, we described that an absence of TET1 resulted in post-gastrulation embryonic defects (Khoueiry et al, 2017; Luo et al, 2020), in agreement with observations by other groups (Fong et al, 2016; Yamaguchi et al, 2013). In a $Tet1^{GT(RRG140)}$ strain expressing the TET1 exon2-β-geo fusion protein, we initially observed gross malformations in mixed B6;129P2/OlaHsd $Tet1^{GT/GT}$ embryos at early backcrosses; intriguingly, $Tet1^{GT/GT}$ embryos were rescued after 6 generations of backcrossing to B6, but succumbed to 100% lethality within two generations of outbreeding to CD1 (Khoueiry et al, 2017). To eliminate the possibility of embryonic lethality due to dominant negative effects of ectopic GT fusion protein expression in $Tet1^{GT/GT}$ mice, we engineered a new gene-targeted B6-$Tet1^{tm1Koh}$ mouse strain, in which an insertion cassette at the first coding ATG effectively abolishes all coding transcripts of full-length *Tet1* expressing an embryonic isoform (Bartoccetti et al, 2020; Khoueiry et al, 2017). In the B6-$Tet1^{tm1Koh}$ strain, we observed a 25% failure in rostral neuropore closure without developmental delay or other gross abnormalities among $Tet1^{-/-}$ (knockout, KO) embryos, indicating a specific impact on cranial NTC, while no NTD was observed in $Tet1^{+/+}$ (wild type, WT) and $Tet1^{+/-}$ (heterozygous, HET) littermates (Fig. 1A).

To demonstrate strain-dependency of NTD phenotypic penetrance in *Tet1* KO, we outcrossed the B6-$Tet1^{tm1Koh}$ congenic strain to genetically heterogeneous outbred CD1 lines for at least three generations. Indeed, NTDs specific to *Tet1* KO embryos were observed at rates reaching 70%, confirming a significant contribution of genetic modifiers to the phenotype, as previously observed in $Tet1^{GT/GT}$ mice (Khoueiry et al, 2017) (Fig. 1A). In NTD-affected CD1-*Tet1* KO embryos examined from E9.5 until E12.5, we observed a progression from failed fusion of the E9.5 cranial neural ridges to a protrusion of brain tissue outside of the cranial vault (exencephaly) by E11.5 (Copp and Greene, 2010) (Fig. 1B). Histologically, we confirmed incomplete midbrain-hindbrain neuropore closure in *Tet1* KO embryos, while the posterior neuropore closed completely by E10.5 (Fig. 1C; Appendix Fig. S1A). Independently of the NTD phenotype, *Tet1* KO embryos displayed significantly reduced crown-rump (CR) length compared to WT and HET embryos (Fig. 1D), otherwise no overt differences were observed between NTD-affected and unaffected (normal) *Tet1* KO E11.5 embryos (Appendix Fig. S1B). When scored at E11.5, WT, HET and KO embryos were obtained from heterozygote intercrosses in the expected Mendelian ratio (Appendix Fig. S1C), suggesting there was no attrition of KO embryos due to earlier embryonic lethality. There was also no indication of developmental delays in NTD-affected KO embryos compared to normal KO based on Theiler staging (Appendix Fig. S1D). Thus, while the absence of TET1 resulted in smaller embryos as also observed in another gene targeted strain of *Tet1* KO (Chrysanthou et al, 2022), the presence of NTDs did not affect developmental progression.

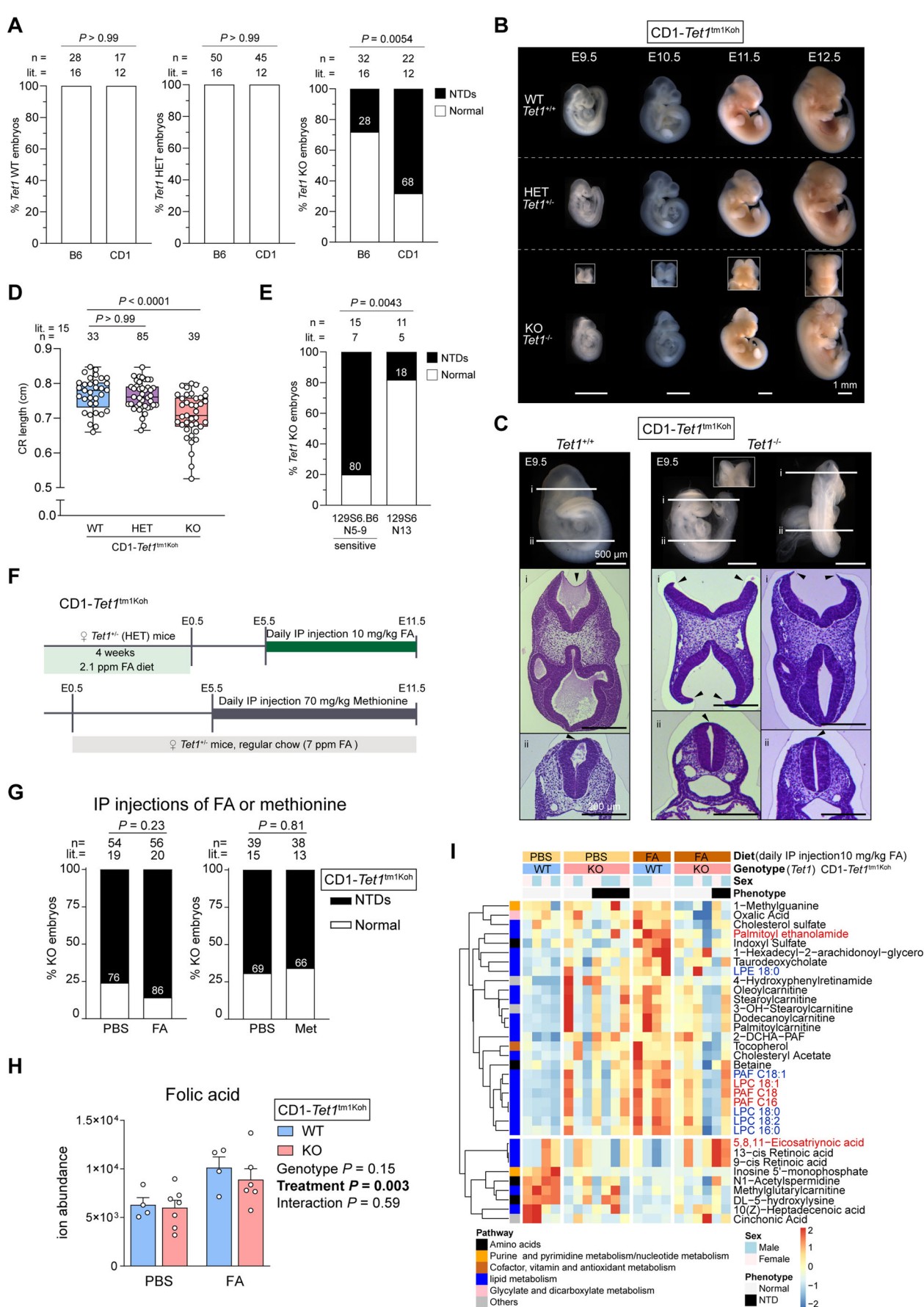

**Figure 1. Phenotypic and metabolomic analysis of strains of Tet1⁻/⁻ mice displaying high or low NTD penetrance and the characterization of strain responsiveness to FA and methionine supplementation.**

(A) NTD rates of $Tet^{+/+}$ (WT), $Tet^{+/-}$ (HET), and $Tet1^{-/-}$ (KO) embryos in B6 and CD1 mouse strains. n = number of embryos, litter (lit.) = number of litters collected. Numbers in black bars are percentage values of NTD cases. Embryos were scored at E10.5-E13.5. P values are calculated by Fisher's Exact test. (B) Images of E9.5 to E11.5 embryos per genotype of CD1-$Tet1^{tm1Koh}$ mice. In $Tet1^{-/-}$ embryos, the failure of neuropore closure is shown with a frontal view image (above) of the forebrain at each stage. Scale bar indicates 1 mm. (C) Hematoxylin & eosin-stained sections of E9.5 CD1-$Tet1^{+/+}$ and $Tet1^{-/-}$ embryos. Horizontal lines in whole embryo images (top) indicate planes of section at cranial (i) and caudal (ii) regions. Arrowheads indicate failed neuropore closure in cranial (i) but normal closure in caudal (ii) sections of $Tet1^{-/-}$ embryos. Scale bar indicates 500 μm in images and 200 μm in sections. (D) Crown-rump length of E11.5 WT, HET, and KO embryos from CD1-$Tet1^{tm1Koh}$ mice. P values are calculated by one-way ANOVA. For simpler visualization, exact P value was not shown when the value < 0.0001. The box plot shows the interquartile range, with the lower edge indicating the 25th percentile and the upper edge the 75th percentile. The line inside the box indicates the median. The minimum value is indicated by the lower whisker, while the maximum value is indicated by the upper whisker. There was no outlier. (E) NTD rates of $Tet1^{-/-}$ embryos upon backcrossing B6-$Tet1^{tm1Koh}$ to a 129S6 strain. P values are calculated by Fisher's Exact test. (F) Schematics of the experimental protocols for FA and methionine supplementation by intraperitoneal (IP) injection in CD1-$Tet1^{tm1Koh}$ heterozygous dams. PBS was injected as vehicle control. (G) NTD rates in CD1-$Tet1^{-/-}$ embryos in response to FA and methionine supplementation. P values are calculated by Fisher's Exact test. (H) HPLC-MS/MS detection of FA in E11.5 CD1-$Tet1^{tm1Koh}$ whole embryos. n = 4, WT embryos in PBS and FA-injected groups; n = 7, KO embryos in PBS-injected; and n = 6, KO embryos in FA-injected groups. Overall P values are calculated by two-way ANOVA to compare the effect of genotype factor, treatment factor, and genotype × treatment interaction, incorporating a post hoc Tukey's test for multiple comparison correction. Error bar represents mean ± standard error of the mean (SEM). (I) Heatmap of HPLC-MS/MS ion abundance of metabolites detected in E11.5 CD1-$Tet1^{tm1Koh}$ whole embryos, filtered for compounds exhibiting significant changes (two-way ANOVA interaction P < 0.05) in response to FA supplementation and $Tet1$ KO. n = 4, WT embryos in PBS and FA-injected groups; n = 7, KO embryos in PBS-injected; and n = 6, KO embryos in FA-injected groups. Phospholipids are highlighted in bold blue, neurotransmitters and metabolites related to neuronal endocytosis/exocytosis in bold red. Source data are available online for this figure.

We investigated whether the high penetrance of NTDs in $Tet1$ KO embryos depends on genetic heterogeneity present in outbred and mixed strains or could be recapitulated in an alternative inbred congenic strain. To explore this, we backcrossed the B6-$Tet1^{tm1Koh}$ allele to an available 129 substrain (129S6, also known as 129/SvEvTac). Interestingly, incipient congenic 129S6.B6-$Tet1$ KO embryos (within 5–9 generations of backcrossing) exhibited a peak in phenotype penetrance as high as 80% (Fig. 1E). Further backcrossing (≥13 generations) to create a highly inbred 129S6 congenic (129S6.Cg) strain reduced NTD rates in $Tet1$ KO to below 20% (Fig. 1E), suggesting that some genetic heterogeneity (in CD1 and 129S6.B6 incipient congenic) is necessary to promote the phenotypic expression of NTDs in $Tet1$ KO embryos.

## NTDs in $Tet1^{-/-}$ embryos are unaffected by FA status across strains

Knowing that FA is a major protection against NTD risk, we asked whether the penetrance of NTDs caused by loss of $Tet1$ is sensitive to maternal folate status across mouse strains. We initially designed experiments to explore the potential of surplus FA to mitigate NTD penetrance in the highly susceptible CD1-$Tet1^{tm1Koh}$ KO mice, and conversely, whether FA deprivation could elevate NTD rates in the less sensitive congenic B6-$Tet1^{tm1Koh}$ KO strain (Fig. 1F; Appendix Fig. S1E). Prior to timed-mating with CD1-$Tet1$ HET studs, CD1-$Tet1$ HET dams were adapted to a custom diet containing 2.1 ppm FA for 4 weeks, reduced from 7 ppm in vitamin-enriched chow regularly used in our mouse facility. Following a previously published study (Burren et al, 2008), we injected the dams with 10 mg/kg FA (or PBS as vehicle control) intraperitoneally daily from E5.5 and collected embryos at E11.5 (Fig. 1F). Direct FA injection did not reduce NTD rates in CD1-$Tet1$ KO embryos compared to controls and might even slightly exacerbate NTD penetrance to as high as 85%, although the difference did not reach statistical significance (Fig. 1G). To address the possibility that the cycling of folate derivatives may be uncoupled with that of methionine (the direct substrate for the generation of SAM), we also tested injections of methionine at 70 mg/kg (Leung et al,

2017) in CD1-$Tet1$ HET dams (Fig. 1F), but this treatment showed no effect on NTD rates in KO offspring (Fig. 1G). To perform FA deprivation in B6 mice, we adapted B6-$Tet1$ HET dams to a diet containing 0.1 ppm FA for 4 weeks before mating with $Tet1$ HET studs. To further eliminate folate synthesis by intestinal microbiota, we supplemented the custom diet with an antibiotic, 1% succinyl sulfathiazole (SST), as used previously (Burren et al, 2008) (Appendix Fig. S1E). The control diets for these sets of experiments contained 2.7-3 ppm FA plus 1% SST. Here as well, FA deficiency did not increase NTD rates in B6-$Tet1$ KO embryos (Appendix Fig. S1F). Interestingly, among B6-$Tet1$ HET embryos in the FA-deprived group, we observed one case of NTD (Appendix Fig. S1F,G). Nonetheless, NTDs caused by loss of TET1 were non-responsive to FA dietary changes in both inbred and outbred mice.

Folate 1C cycling provides substrates for methylation reactions, but TET dioxygenases erase DNA methylation to counteract excessive DNMT activities. Thus, loss of TET1 activity may impact 1C metabolism in embryos even if resulting NTDs were non-responsive to FA supplementation. To gain a comprehensive view of steady-state metabolic changes induced in $Tet1$ WT and KO embryos in response to maternal FA dietary status, we performed an untargeted high-performance liquid chromatography with tandem mass spectrometry (HPLC-MS/MS) analysis of 598 metabolites detectable in E11.5 whole embryos. Notably, FA administration to CD1-$Tet1^{tm1Koh}$ WT embryos resulted in a significant increase in FA and a global increase in metabolites belonging to phospholipid and carnitine pathways, a response not mirrored to the same extent in $Tet1$ KO embryos (Fig. 1H,I). Conversely, FA deprivation in B6-$Tet1^{tm1Koh}$ WT embryos resulted in downregulation of phospholipids, but an increase in carnitine metabolites, while $Tet1$ KO embryos remained non-responsive (Appendix Fig. S1H). These differences in metabolomic response between WT and $Tet1$ KO embryos were observed independently of embryo sex and NTD presentation, suggesting that the loss of $Tet1$ is the dominant factor disrupting the homeostatic regulation of 1C metabolism in response to nutritional insults post-neurulation. Comparing the metabolomic response between the two studies

conducted in CD1 and B6 mice, we conclude that carnitine metabolism in WT embryos was affected in response to either an excess or depletion of FA, and in both cases the response was abolished by a loss of TET1. On the other hand, the levels of phospholipid metabolites appear positively correlated with FA status, in line with folate being a source of 1 C for lipid metabolism (da Silva et al, 2014), and changes were again lost in the absence of *Tet1*. In contrast, injections of methionine did not produce any significant changes in methionine and FA-responsive metabolites detectable in the embryos (Appendix Fig. S1I,J).

## Maternal folate status and *Tet1* gene dosage contribute synergistically to the incidences of congenital brain malformations

To further investigate whether adverse maternal dietary folate status modifies the impact of *Tet1* dysfunction on offspring development, we re-designed our experiments to adapt CD1-*Tet1* HET dams to custom diets containing modified FA levels. According to "US FDA 2005 Guideline for Industry" to convert doses in human versus in a small animal, an adult mouse consuming 3–5 g chow daily on a 3 ppm FA diet has an intake like 0.11–0.18 mg of FA per day in human (US CDC recommended dietary allowance is 0.4 mg). On this basis, a high FA diet containing 30 ppm FA was designed to achieve a 10-fold FA excess in rodents comparable to >1 mg daily intake in human (Cao et al, 2023). To establish a well-defined level of maternal folate availability, custom diets were supplemented with 1% SST, as described above (Appendix Fig. S1E). In line with previous studies (Burren et al, 2008; Christensen et al, 2015; Cosin-Tomas et al, 2020), the scientific rationale for adding SST is to establish an accurate/known level of FA availability from the diet and to reduce experimental variability, by excluding the microbiota as a major source of folate in rodents. Using CD1-*Tet1*[tm1Koh] mice, we verified that NTD rates of *Tet1* KO were not significantly altered by custom diets containing 2.7 or 7 ppm FA (matching levels in regular chow) with or without SST (Appendix Fig. S2A).

Custom diets containing 30 ppm (excess), 3 ppm (control), or 0.1 ppm (depleted) FA (with 1% SST) were administered to timed-pregnant CD1-*Tet1* HET dams, and embryos were phenotyped as before at E11.5 (Fig. 2A). As reference, an additional group (regular chow) was kept on standard chow enriched with 7 ppm FA (without SST) regularly used in our animal facility. Again, we observed that modified FA levels in the custom diets did not significantly alter NTD penetrance rates among CD1-*Tet1* KO embryos; there was even a (non-significant) trend of increasing NTD penetrance among KO embryos with increasing FA levels in the experimental cohort (Fig. 2B). While all *Tet1* WT embryos were normal, mild morphological abnormalities, particularly at the midbrain-hindbrain junctures, were observed in 3% of HET and ~5% of KO embryos upon exposure to excess or depleted FA (Fig. 2B; Appendix Fig. S2B), suggesting that these cranial "closed neural tube" structural defects may arise synergistically from the combined loss of *Tet1* gene copy and adverse levels of FA. In agreement, neurodevelopmental deficits have previously been reported in rodent offspring exposed to adverse (excess and deficient) maternal folate status (Harlan De Crescenzo et al, 2021; Wang et al, 2021).

The genetic heterogeneity of CD1 outbred mice could be a source of high variance in NTD penetrance (ranging from 50 to 70%, Fig. 2B) observed between different cohorts of *Tet1* KO embryos. To eliminate this confounder, we performed subsequent experiments using the 129S6 congenic (129S6.Cg) *Tet1*[tm1Koh] strain of mice after 13 generations of backcrossing to 129S6. Based on the higher NTD penetrance observed in incipient congenic 129S6.B6-*Tet1* KO embryos, we reason that the 129S6 background would contain the bulk of genetic variants that increase susceptibilities for developing embryonic defects compared to B6. The low NTD penetrance in 129S6.Cg-*Tet1* KO mice would also facilitate the detection of other structural defects at E11.5 post-neurulation. In the 129S6.Cg strain, the use of custom diet introduced a basal rate of gross malformation (~5% to 13%) and increased resorption rates especially in low FA condition (Fig. 2C,D; Appendix Fig. S2C,D). Grossly malformed phenotypes were specifically associated with an exposure to SST in the custom diets (not observed in regular chow groups) and can be regarded as a side-effect of antibiotic treatment unrelated to FA status. Nonetheless, we observed an increased incidence of morphological abnormalities at the hindbrain-midbrain junctures in *Tet1* WT and HET embryos exposed to FA excess or deprivation, which was much more prominent in the 129S6.Cg strain than in CD1 outbred (Fig. 2B,C). The penetrance of such "brain malformation" was notably exacerbated among 129S6.Cg HET embryos exposed to excess and depleted FA (20–30%, compared to 10–15% among WT embryos), significantly above the basal rates among HET embryos (<5%) in the control and regular chow groups. These results suggest that a coupling of adverse maternal FA with *Tet1* haploinsufficiency may have subtle effects on embryonic brain development exacerbated in inbred congenic strains. Like in CD1 mice, open NTDs in 129S6.Cg *Tet1* KO embryos were observed at similar rates under all dietary conditions (Fig. 2C). Strikingly, we found 3 NTD cases in two litters among 46 HET embryos examined (6.5%), specifically in the FA depleted groups. Previously, NTDs were never observed in HET embryos under standard chow conditions (in over 1000 experimental time-pregnancies conducted using B6, 129S6 and CD1 strains of *Tet1*[tm1Koh] mice). Together with a case of NTD also observed among FA-deprived B6-*Tet1* KO embryos (Appendix Fig. S1F), these data suggest that *Tet1* haploinsufficiency can contribute to increased susceptibility to NTDs and other congenital malformations under conditions of maternal FA deficiency.

## Adverse maternal folate status coupled with loss of *Tet1* disrupts phospholipid metabolism

To understand how maternal FA intake and *Tet1* gene dosage interact to shape cellular physiology, we proceeded to dissect molecular changes at the level of the metabolome, DNA methylome, and transcriptome. To profile the metabolomic changes, we again performed untargeted HPLC-MS/MS in whole E11.5 129S6.Cg-*Tet1*[tm1] embryos per custom diet and *Tet1* genotype (3 × 3 experimental groups), detecting 807 metabolites. In each biological replicate ($n = 6$), KO embryos were stage- and sex-matched to littermate WT and/or HET embryos, excluding embryos with gross malformation and developmental delays (Fig. 3A). In our sample sets comprising 54 embryos (Table EV1), we intentionally included a few WT and HET embryos with mild deformities (but developmentally matched by Theiler stages) to ensure that the sampling is representative of the range of phenotypes observed. By the same criteria, we excluded KO

**A**

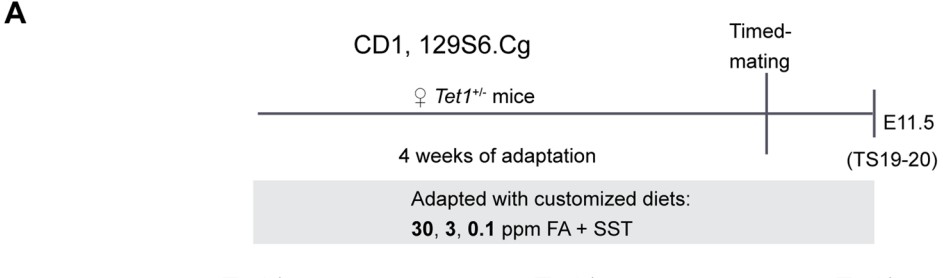

**B**

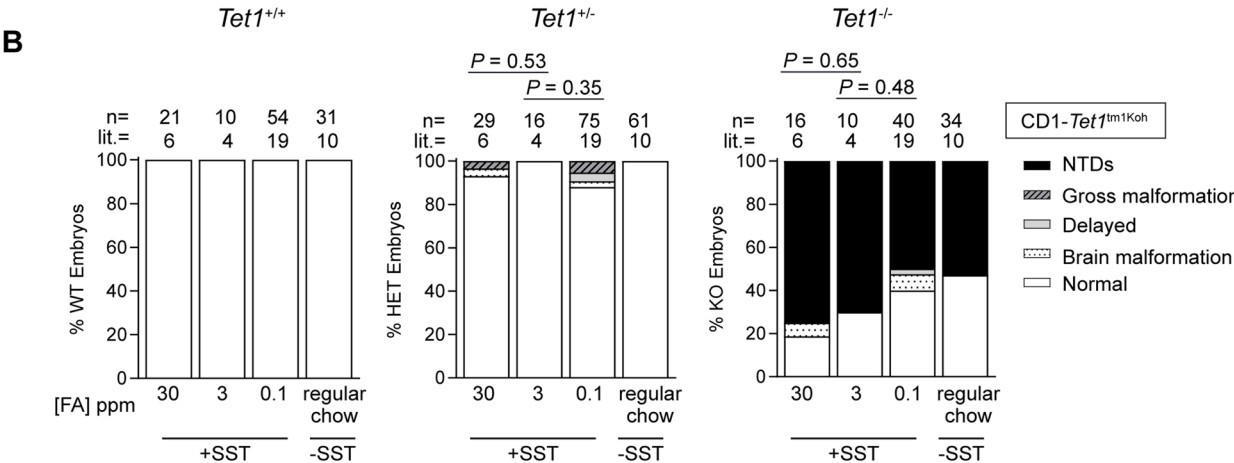

**C**

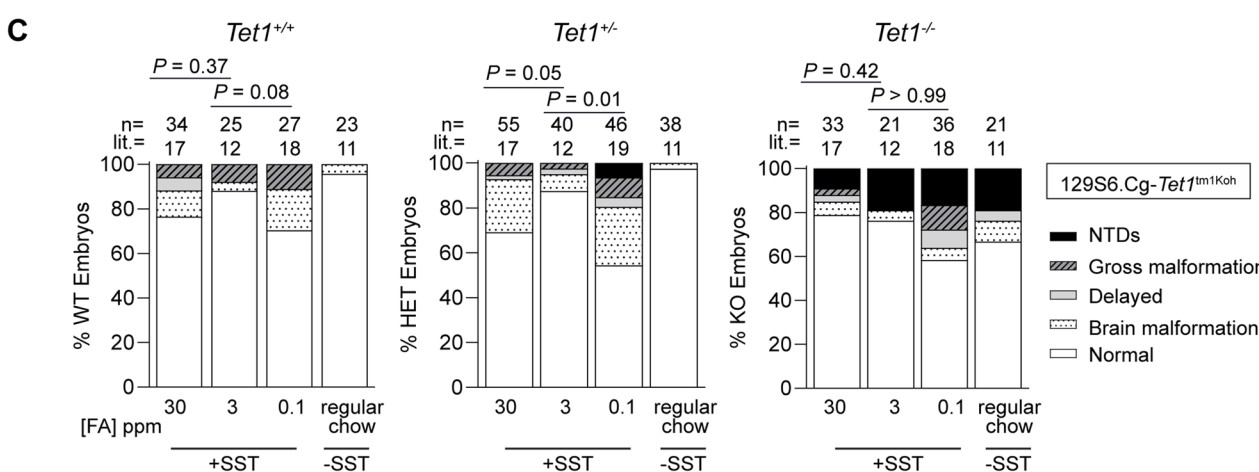

**D**

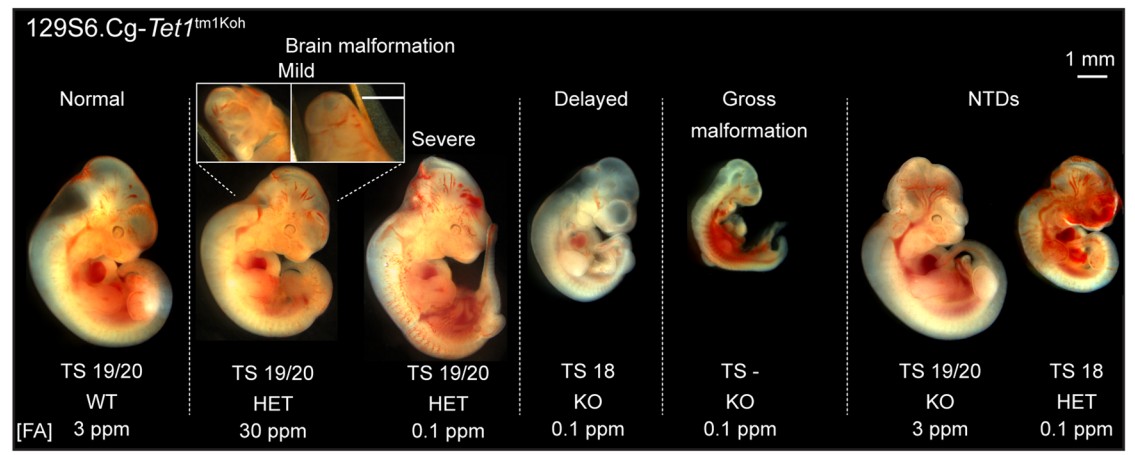

◄ **Figure 2. Phenotypic analysis of CD1 and 129S6.Cg E11.5 embryos as an effect of *Tet1* genotype and maternal dietary FA excess or depletion.**

(A) Schematic of protocol adapting *Tet1*$^{+/-}$ dams to custom diet containing excess (30 ppm), normal (3 ppm, control) or depleted (0.1 ppm) FA in CD1 and 129S6.Cg backgrounds. 30 ppm, 3 ppm, and 0.1 ppm FA custom diets were made based on the AIN-93G semi-purified diet formulation (TestDiet® 57W5) except for FA concentrations. All three diets contain the antibiotic, succinyl sulfathiazole (SST). (B) Phenotypic scoring of *Tet1*$^{+/+}$, *Tet1*$^{+/-}$ and *Tet1*$^{-/-}$ embryos from CD1-*Tet1*$^{tm1Koh}$ mice fed custom modified FA diets. Regular chow contains 7 ppm FA without SST. *P* values are calculated by Fisher's Exact test. In the *Tet1*$^{+/-}$ genotype group, *P* values indicate the statistical difference of all affected embryos compared to normal embryos; in *Tet1*$^{-/-}$, *P* values indicate the statistical difference of embryos with NTD compared to normal embryos. (C) Phenotypic scoring of *Tet1*$^{+/+}$, *Tet1*$^{+/-}$ and *Tet1*$^{-/-}$ embryos from the 129S6.Cg-*Tet1*$^{tm1Koh}$ strain fed custom FA diets or regular chow as in (B). *P* values are calculated by Fisher's Exact test. In the *Tet1*$^{+/+}$ and *Tet1*$^{+/-}$ genotype groups, *P* values indicate the statistical difference of embryos with brain malformation compared to normal embryos; in the *Tet1*$^{-/-}$, *P* values indicate the statistical difference of embryos with NTD compared to normal embryos. (D) Representative images of E11.5 embryos from the 129S6.Cg-*Tet1*$^{tm1Koh}$ strain, illustrating the range of distinct phenotypes associated with custom modified FA diet and *Tet1* genotype as in (C). The closure of neuropore in the brain malformation category is shown with both front and back views. Theiler stage, *Tet1* genotype and diet are indicated below each embryo. Scale bar is 1 mm. Source data are available online for this figure.

embryos with severe malformation when they could no longer be stage-matched with WT controls, minimizing confounding DNA methylation changes due to gross developmental delays.

Focusing first on metabolites in the folate and methionine cycles (Fig. 3B), we verified whether the modified maternal FA diet had a significant impact on their levels in the embryos per *Tet1* genotype. We observed an increase in detectable FA in WT and HET embryos given excess FA, but this increase was nullified in KO embryos (Fig. 3C). Two-way ANOVA indicated an overall significant impact on detectable FA by both diet and genotype factors (independently of each other) in pairwise comparisons among all groups. Increased FA levels in WT and HET embryos under excess FA supplementation was not associated with any significant increase in methionine or decrease in homocysteine, suggesting that not all ingested FA had been metabolized (Stam et al, 2005) (Appendix Fig. S3A,B). In KO embryos supplemented with excess FA, methionine abundance was significantly reduced compared to the levels in WT embryos (Appendix Fig. S3A), in line with reduced levels of FA intake. Although changes due to genotype-diet interactions were not significant by two-way ANOVA, the progressive decrease in SAM/SAH (a measure of cellular methylation potential) and increase in serine/glycine ratios (indicative of reduced rates in cycling of active folate species) with reducing FA levels were accentuated by loss of *Tet1* (Fig. 3D,E). These findings indicate a sensitivity of folate/methionine cycling towards *Tet1* gene dosage under conditions of either excess or depleted FA.

Principal component analysis (PCA) of 84 metabolites exhibiting significant changes in ion abundances in at least one experimental group using one-way ANOVA showed sample groups clustering by dietary FA status along the first principal component (PC1). As an exception, KO embryos exposed to excess FA (group KO-30 ppm FA) distinctly separated from HET and WT embryos along both PC1 and PC2 (Fig. 3F). PCA of the differential metabolome in individual embryos showed two (TP12.3_3ppm_HET_female and TP25.6_30ppm_WT_male) to be outliers from replicates in their respective groups (Appendix Fig. S3C). Noting smaller embryo size relative to littermates (TP25.6_30ppm_WT_male) and mild abnormalities in embryo morphology (TP12.3_3ppm_HET_female), we excluded these two samples in downstream analysis. To comprehend the metabolomic changes resulting from an interaction of both FA diet and *Tet1* genotype, we filtered the significant metabolites further using two-way ANOVA. This analysis revealed that a majority (40.4%, $n = 21$) of metabolites significantly affected by the diet-genotype interaction (total $n = 52$) belonged to pathways of lipid metabolism (Fig. 3G; Appendix Fig. S3D). Specifically, several clusters, including

phosphatidylcholine (PC) metabolites and neurotransmitters, were significantly increased in WT and HET embryos treated with excess FA, but not in KO (Fig. 3G). Conversely, these PC compounds were reduced in WT and HET embryos relative to KO upon FA depletion, strongly supporting that the positive correlation between FA and PC levels is uncoupled by the loss of TET1. The differences were observed independently of the embryo's sex and phenotype and recapitulated the compromised metabolomic response of *Tet1*$^{-/-}$ embryos to adverse folate status in CD1 and B6 mice (Fig. 1I; Appendix Figs. S1H and S3D). Together, these findings underscore an intricate functional interaction between TET1 and folate 1C metabolism and suggest that FA-resistant NTDs observed in *Tet1* null embryos may stem from an impaired ability to metabolize or uptake FA.

## FA-sensitive differentially methylated regions in *Tet1* KO are enriched for neurodevelopmental loci

To discern the consequences of *Tet1* loss and dietary variations in FA on cellular methylation reactions, we profiled genome-wide DNA methylation patterns as an epigenetic readout in 129S6.Cg embryonic brains exposed to all three FA diets. For this genomic analysis, we dissected E11.5 brain tissues from the most rostral aspect of the forebrain to the caudal aspect of the hindbrain above the otic vesicle, following ENCODE guidelines (He et al, 2020) (Appendix Fig. S4A). We selected individual embryos matched by somite counts, pairing a KO with one WT and/or HET of the same sex within a litter as a replicate unit (Fig. 3A). Our final sample set included a total of 36 embryos collected from four litters in each of the three diet groups, representative of embryo phenotypes and both sexes (Table EV2). To profile DNA methylation changes in this sample cohort cost-effectively, we performed enhanced reduced representation bisulfite sequencing (RRBS), which is based on enrichment of MspI and TaqI restriction sites (Meissner et al, 2008). From sequencing 20 million PE150 reads per sample, we obtained a mean sequencing depth of 8.7x covering about 2 million CpGs. The PCA projection and cluster analysis showed samples separating into two clusters along PC1 according to KO versus WT/HET genotype, indicating that complete loss of TET1 function, rather than diet, was the dominant driver of methylation differences (Appendix Fig. S4B,C). One distinct outlier sample (HET-0.1 ppm FA), coincidentally also the sample with the lowest coverage of 4.9x, was excluded in downstream analysis (Appendix Fig. S4B).

Aggregate methylation level profiles of individual CpGs across the gene structure of 21231 protein-coding genes, when stratified

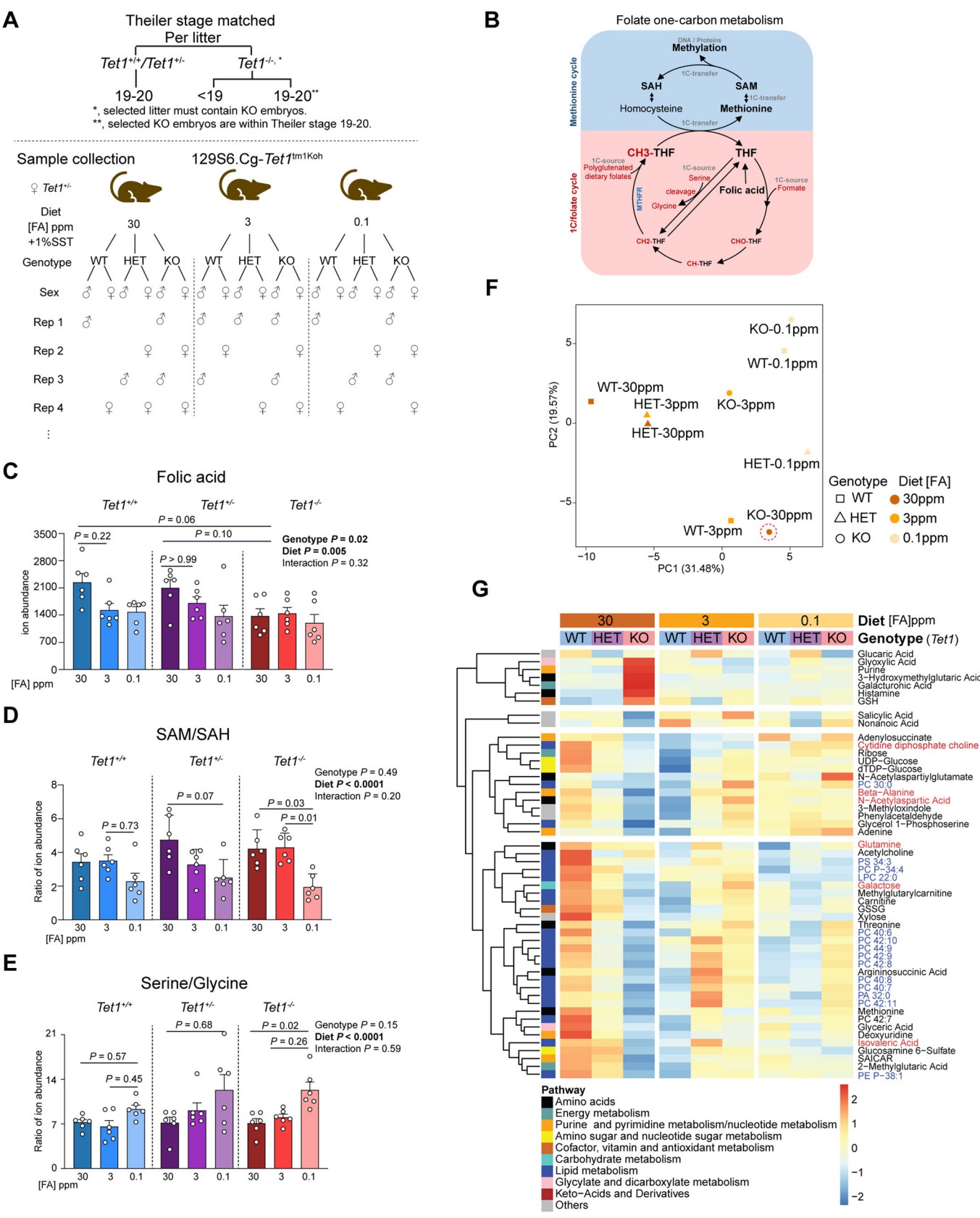

◄ **Figure 3.  Metabolomic changes in 129S6.Cg embryos as an effect of *Tet1* genotype and modified maternal dietary FA status.**

(A) Schematic of sample selection criteria to match KO with littermate WT and/or HET embryos by Theiler stage (TS19-20) and sex per biological replicate (rep) at the experimental endpoint of E11.5. (B) Schematic diagram of folate one-carbon cycling to methylation reactions. THF, tetrahydrofolate; SAH, S-adenosyl-L-homocysteine; MTHFR, methylenetetrahydrofolate reductase. (C–E) HPLC-MS/MS detection of FA (C), SAM/SAH (D), and serine/glycine (E) ratios in whole embryos. Data are mean ± SEM from $n = 6$ individual embryos, 3 males and 3 females per group, except WT-30 ppm FA and KO-3 ppm FA groups which consisted of 4 females and 2 males each. Overall P values are calculated by two-way ANOVA and a post hoc Tukey's test for multiple comparison correction. P values shown in pairwise comparisons are determined by one-way ANOVA and a post hoc Dunn's multiple comparison test. For simpler visualization, exact P value was not shown when the value < 0.0001. (F) Principal component analysis (PCA) of HPLC-MS/MS ion abundance metabolomic data of 3 × 3 groups of WT, HET, and KO samples exposed to the three FA diets. (G) Heatmap of metabolites with significant changes resulting from interactions between FA status and *Tet1* genotype (2-way ANOVA interaction $P < 0.05$). Row dendrogram on the left shows k-means clustering of metabolite compounds. Heatmap columns show scaled Z-score of mean ion abundance values of samples within each group. Refer to Appendix Fig. S3C,D for PCA and heatmap respectively of individual embryo samples. $n = 6$ per group, except two groups—HET-3 ppm FA and WT-30 ppm FA—in which $n = 5$ after one outlier was excluded from downstream differential analysis. Phospholipids are highlighted in bold blue, neurotransmitters and metabolites involved in neuronal endocytosis/exocytosis in bold red. Source data are available online for this figure.

by *Tet1* genotypes and FA diets, revealed a common increase in CpG methylation over 5-kb upstream and downstream regions flanking transcription start sites (TSS) and within the gene bodies in KO samples across all FA diet groups (Fig. 4A; Appendix Fig. S4D). While the meta-profiles of WT and HET samples overlapped closely in excess and control FA diet groups, the profile for HET was clearly intermediate between those of WT and KO in the FA depleted group, indicating an effect of *Tet1* haploinsufficiency on increasing methylation exclusively under FA deficiency (Fig. 4A). We further plotted the distribution of all 1 kb tiles by methylation levels, confirming that DNA hypermethylation relative to WT samples occurred in all KO in all three FA diet groups and that an intermediate increase was also evident in HET samples in the FA depleted group (Fig. 4B). To identify differentially methylated regions (DMRs) resulting from differences in *Tet1* genotypes, we then conducted pairwise comparison analysis between KO or HET vs WT, per diet group. As expected, a majority (99.3%) of 33851 DMRs exhibited gains in DNA methylation (hyper DMRs) in KO versus WT across all diets (Fig. 4C). However, excess and depleted FA diets resulted in fewer hyper DMRs compared to the control diet, suggesting that adverse maternal folate levels combined with *Tet1* KO may affect DNA hypermethylation differently than *Tet1* KO alone. Similarly, in the comparison between HET and WT in control FA diet, we also detected 6030 hyper DMRs, indicative of an impact of *Tet1* haploinsufficiency on DNA methylation, which were completely obliterated by modified FA levels. When comparing the two hyper DMR subsets in KO or HET vs WT in the control diet group, we identified 3759 significantly overlapping regions, predominantly enriched in CpG islands (CpGi) and shores associated with promoter regions (Appendix Fig. S5A,B). These 3759 overlapping hyper DMRs were enriched for neuroectoderm ("regulation of neurogenesis", "axonogenesis", "epithelial tube morphogenesis") and mesoderm ("mesenchyme development", "mesenchymal cell differentiation") related gene ontology (GO) terms, suggesting that DNA methylation patterns in multiple cell lineages of the embryonic brain are sensitive to *Tet1* gene dosage independently of FA levels (Appendix Fig. S5C).

Collating all hyper DMRs resulting from *Tet1* KO vs WT comparisons in the three custom FA diet groups, we asked whether we could distinguish regions affected solely by loss of *Tet1* independently of the FA diet, from those affected by both *Tet1* genotype and FA diet. The Venn overlap identified 4386 hyper DMRs common in all three FA diet conditions, which we call

"TET1-common" (Fig. 4D). A corollary of this classification is that hyper DMRs outside of any overlap are dependent on both *Tet1* and each specific FA diet, which we call "TET1 x FA diet-specific". In either category, hyper DMRs were observed independently of the embryo's sex and phenotype (Appendix Fig. S5D). We performed 2-way ANOVA to compare the methylation levels calculated for each subset of hyper DMRs between groups using genotype and diet as independent factors (Appendix Fig. S5E). For TET1-common hyper DMRs, *Tet1* genotype was an extremely significant factor accounting for methylation differences, but diet and the interaction between genotype and diet were not. On the other hand, for TET1 x FA diet-specific DMRs, the interaction between genotype and diet reached significance in addition to genotype alone, indicating that FA diet alone has limited effects on DNA hypermethylation but can couple with the loss of TET1 to impact DNA methylation changes. When annotated by genomic features, TET1-common hyper DMRs were strongly enriched in CpGi and CpG shores, closely associated with gene-proximal promoter regions, while TET1x FA diet-specific hyper DMRs were associated more with distal loci to a level comparable with the coverage of the genome (Fig. 4E). Strikingly, GO terms associated with TET1-common DMRs were mainly mesoderm-related, including "muscle system process", and "muscle contraction" and "muscle tissue development" (Fig. 4F). In contrast, GO terms associated with TET1x FA diet-specific DMRs were mostly ectoderm-related terms, including "axonogenesis", "epithelial tube morphogenesis", "regulation of neurogenesis", "neuron projection guidance", and "axon guidance" (Fig. 4F). Collectively, these analyses suggest that the two classes of TET1-common and TET1 x FA diet-specific hyper DMRs are derived from biological perturbations affecting distinct genomic loci. Specifically, TET1-common hyper DMRs, resulting from the loss of TET1 as a primary driver of DNA hypermethylation independently of diet, affected mainly mesodermal gene promoters. In contrast, TET1 x FA diet-specific hyper DMRs, resulting from an interaction between FA diet and *Tet1* genotype, affected both promoter proximal and distal loci that regulate neurodevelopmental functions.

When we compared excess or depleted FA with control FA diet conditions to assess the dietary impact on each genotype, we discovered >10-fold fewer DMRs, the majority (93.6% of a total of 7057) afflicting WT embryos in both excess and depleted FA diet groups (Fig. 4G). Excess FA induced predominantly hyper DMRs (65.3%) in WT embryos, consistent with the expected positive influence of folate cycling on DNA methylation pathways. The 857

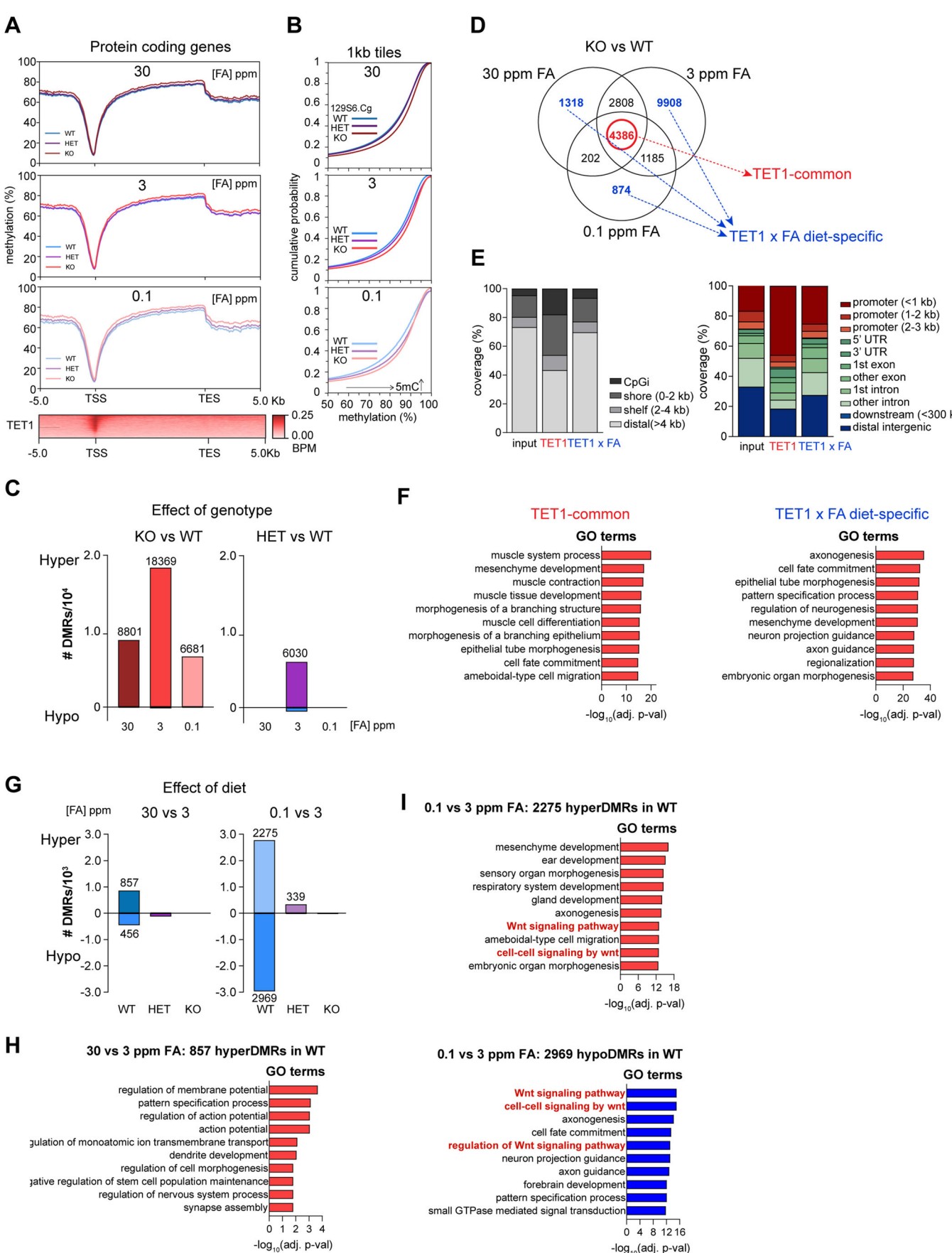

**Figure 4. DNA methylation changes in E11.5 129S6.Cg embryonic brains as an effect of *Tet1* genotype and modified maternal dietary FA status.**

(A) RRBS profiles of aggregate methylation levels across the gene structure of 21231 protein coding genes in E11.5 brain tissues collected from WT, HET, and KO embryos in each FA diet group. $n = 5$, KO per diet group, matched to $n = 3$ WT and $n = 6$ HET in 30 ppm FA, $n = 4$ WT and $n = 3$ HET in 3 ppm FA, and $n = 2$ WT and $n = 3$ HET in 0.1 ppm FA. Bottom, compressed heatmap of TET1 ChIP-seq signals in normalized bins per million mapped reads from 5 kb upstream of transcription start sites (TSS) to 5 kb downstream of transcription termination sites (TES) of these genes. (B) Cumulative distribution plot of methylation levels over 1 kb tiles across the genome in WT, HET, and KO embryos in each FA diet group. Black arrow denote a shift in the distribution curve towards the right with hypermethylation. (C) Number of hyper and hypo differential methylated regions (DMRs) defined by pairwise comparisons between *Tet1* genotypes (KO or HET vs WT control) per diet group. (D) Venn diagram of all hyper DMRs defined by KO vs WT pairwise comparisons per diet. DMRs in the overlap of all three diets are circled in red and called TET1-common, whereas the collective set of DMRs in the non-overlap are colored in blue and called FA x TET1 diet-specific. (E) Distribution by CpG island (CpGi) proximity (left) and gene feature annotation (right) of TET1-common and FA x TET1 diet-specific DMRs. (F) Top 10 GO terms associated with TET1-common (left), and FA x TET1 (right) DMRs. (G) Number of hyper and hypo differential methylated regions (DMRs) defined by pairwise comparisons between FA diet groups (30 ppm or 0.1 ppm vs 3 ppm control) per *Tet1* genotype. (H) Top 10 GO terms associated with 857 hyper DMRs identified from the pairwise comparison of 30 vs 3 ppm FA in WT genotype. (I) Top 10 GO terms associated with 2275 hyper DMRs (left) and 2969 hypo DMRs (right) identified from the pairwise comparisons of 0.1 vs 3 ppm FA in WT genotype. In F, H and I, a hypergeometric test was applied for the enrichment analysis, followed by Benjamini-Hochberg (FDR correction) to adjust the *P* values. FDR: false discovery rate.

hyper DMRs were dramatically enriched in CpGi promoter regions and associated with GO terms primarily related to nervous system development and function, including "regulation of membrane potential", "regulation of action potential", "dendrite development" and "synapse assembly" (Fig. 4H; Appendix Fig. S5F). In contrast, FA depletion induced gain and loss in methylation equally, with DMRs outnumbering those caused by excessive FA, suggesting that FA deprivation has a more extensive impact on the epigenome than FA excess (Fig. 4G). Both hyper DMRs and hypo DMRs induced by FA deficiency in WT embryos were associated with common GO terms for development, most notably "axonogenesis", "Wnt signaling pathway" and "cell-cell signaling by Wnt" (Fig. 4I). Especially, Wnt signaling terms were the top GO terms enriched among hypo DMRs, together with other terms related to neurodevelopment including "neuron projection guidance" and "forebrain development." On the other hand, hyper DMRs induced by FA deficiency enriched for GO terms related to development of broader tissue types ("ear", "respiratory system", and "gland"). When annotated by genomic features, hyper DMRs induced by FA deficiency in WT embryos were relatively more enriched in distal intergenic regions, while hypo DMRs enriched in promoter regions (Appendix Fig. S5G). Thus, the functional impact of FA deficiency may be more directly inducing loss of CpG methylation at gene promoters regulating brain development.

To further validate our RRBS analysis, we selected one WT and one KO sample from both excess FA and control diet groups for whole-genome bisulfite sequencing (WGBS). Compared to the WT samples, we observed marginally increased global CpG methylation levels in the KO samples, but similar overall methylation levels in the two diet groups (Appendix Fig. S6A). To identify DMRs, we conducted pairwise comparisons of either KO versus WT in the two diets, or 30 ppm versus 3 ppm FA in the two genotypes. Similarly to the RRBS datasets, a 10-fold higher number of DMRs was detected by WGBS from the loss of *Tet1* than from the diet variation alone, and a majority (>99%) were hypermethylated in *Tet1* KO (Appendix Fig. S6B). However, unlike the RRBS analysis, an equal number of hyper DMRs were detected in KO from the two diet groups. When interrogating the impact of excess FA on WT and KO genotype, there were slightly more hyper DMRs in the KO sample (139 in KO, 79 in WT). Annotating the two sets of *Tet1* KO hyper DMRs per diet by genomic features, we noted a slightly

stronger enrichment of KO hyper DMRs from control diet within CpGi, shores and shelf promoter regions than those from excess FA diet (Appendix Fig. S6C). These findings suggest that excess FA may be causing hyper DMRs in *Tet1* KO to shift from promoter proximal to more distal regions. Because distal sites are typically of lower CpG density (Weber et al, 2007) and poorly covered by RRBS, this could account for the reduction in RRBS-detectable hyper DMRs in *Tet1* KO brains when exposed to excess FA (Fig. 4C).

Comparing the two sets of KO vs WT hyper DMRs obtained in excess and control FA diets, we identified 1935 overlapping regions (Appendix Fig. S6D). These are mostly enriched for broad developmental GO terms including "regulation of membrane potential", "regulation of nervous system process", "regulation of synapse structure or activity" and "heart contraction" (Appendix Fig. S6E). The 2460 non-overlapping hyper DMRs in *Tet1* KO treated with excess FA were primarily related to neurodevelopment terms (including "axonogenesis" and "regulation of neurogenesis") (Appendix Fig. S6F), mirroring the profile of TET1 x FA diet-specific DMRs from RRBS (Fig. 4F). On the other hand, GO analysis of the 2483 hyper DMRs in *Tet1* KO from the control FA diet group showed an enrichment in mesoderm-related terms ("muscle organ development", "neuromuscular process") (Appendix Fig. S6G), more similar to the profile of TET1-common regions from RRBS (Fig. 4F). On the whole, other than differences due to genome coverage, both RRBS and WGBS analysis confirmed that excess FA modifies the DNA hypermethylation response in *Tet1* KO embryonic brains to affect primarily neurodevelopmental genes.

## Excess FA coupled with TET1 loss mimics FA deficiency in downregulating membrane transporter expression

To bridge the observed metabolomic and DNA methylation shifts described above with gene expression, we performed bulk mRNA-seq on E11.5 brains collected from 129S6.Cg mice exposed to the three custom FA diets. Again, 4–6 biological replicates (comprising male and female littermate pairings) were collected by sex- and stage-matching KO embryos to littermate WT and/or HET embryos, to compile a total of 40 samples for analysis (Table EV3). PCA projection showed that all samples clustered close together independently of diet group and genotype, except two samples (one

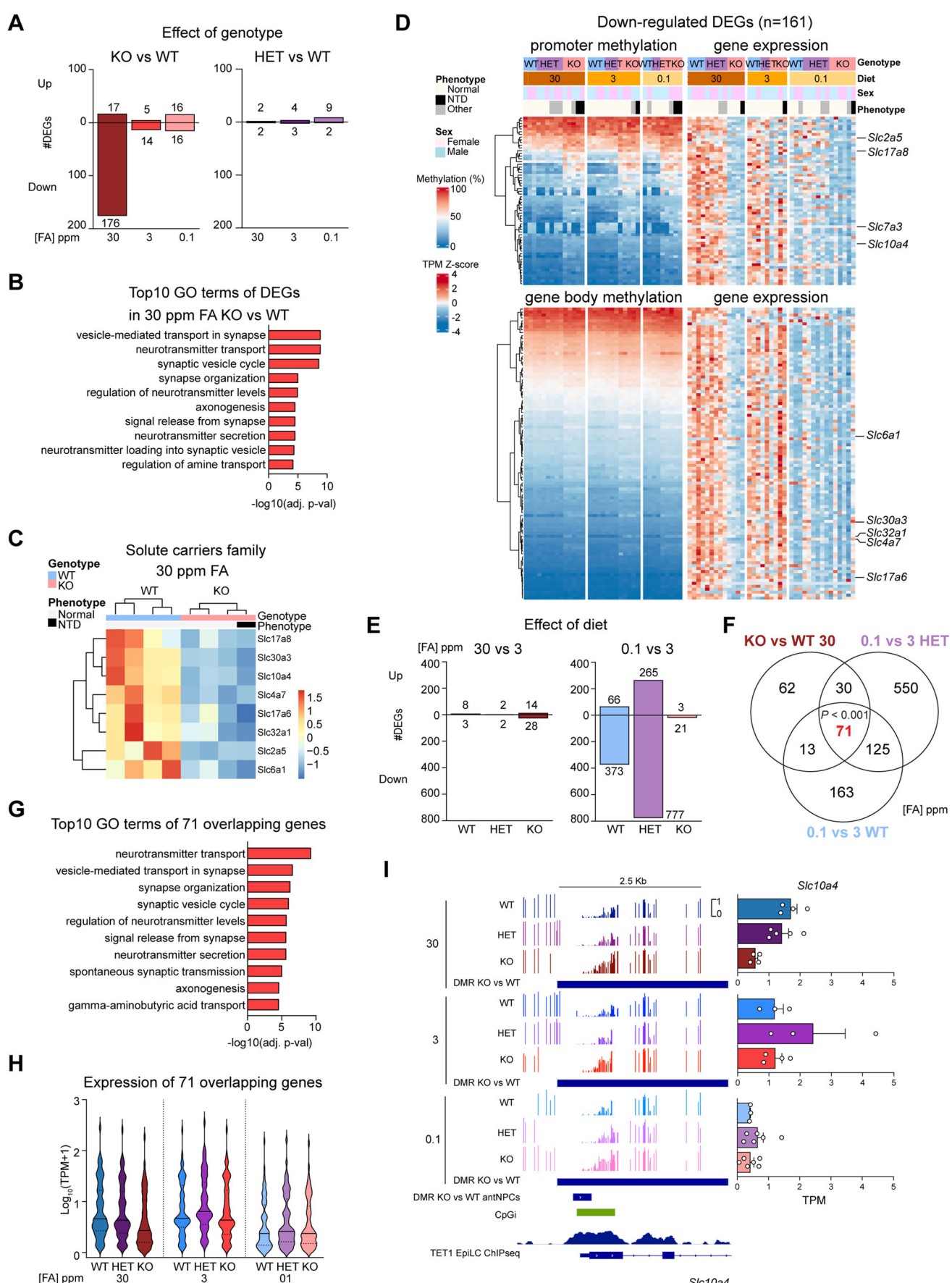

◄ **Figure 5. Convergent loss of solute carrier gene expression in E11.5 embryonic brains caused by either excess FA in the absence of TET1 or by FA depletion.**

(A) Number of RNA-seq differential expressed genes (DEGs) defined by pairwise comparisons between *Tet1* genotypes (KO or HET vs WT), classified by up or down-regulation, in E11.5 embryonic brain samples per diet. The Wald test was used for differential expression testing, with an FDR-adjusted *P* value < 0.05 set as the differential cut-off. (B) Top 10 GO terms associated with the 176 downregulated DEGs from KO vs WT comparison in the 30 ppm FA diet group shown in (A). A hypergeometric test was applied for the enrichment analysis, followed by Benjamini-Hochberg (FDR correction) to adjust the *P* values. (C) Heatmap of RNA-seq expression of DEGs constituting solute carrier family genes in KO versus WT in the 30 ppm FA diet group. Scaled expression is shown as the Z-score of normalized transcripts per million (TPM) values. (D) Expanded heatmap of promoter and gene body methylation levels determined by RRBS correlating with gene expression levels of 161 downregulated DEGs from (A) with sufficient methylation coverage, showing individual embryo per genotype-diet group and annotations of phenotype and sex. The genes are classified by methylation differences between KO and WT greater or less than 5%. SLC family genes are marked on the right. (E) Number of up- and down-regulated DEGs defined by pairwise comparison between diet groups (30 ppm or 0.1 ppm vs 3 ppm FA) per genotype. The Wald test was used for differential expression testing, with an FDR-adjusted *P* value < 0.05 set as the differential cut-off. (F) Venn diagram of all downregulated DEGs defined by KO vs WT in 30 ppm FA in (A), 0.1 ppm vs 3 ppm in WT and HET groups in (E). The number of genes in the overlap is highlighted in bold red. *P* value shown in the overlap is calculated with one-sample binomial test to compare the observed proportion of DEGs in the overlap with the expected proportion under independence of the dietary conditions. For simpler visualization, exact *P* value was not shown when the value < 0.001. (G) Top 10 GO terms associated with the 71 down-regulated DEGs in the overlap defined in (F). (H) Violin plots for the normalized expression levels of the 71 overlapping genes in WT, HET and KO stratified by diets. (I) Integrative Genomics Viewer (IGV) snapshots of RRBS CpG methylation levels over a DMR at the *Slc10a4* gene locus in each genotype-diet group (left). Locations of a DMR identified from WGBS of KO vs WT anterior neuronal progenitor cells (antNPCs) derived by directed in vitro differentiation from ESCs (Data ref: van der Veer et al, 2023), CpG island (CpGi) annotation and TET1 ChIP-seq signals in epiblast-like cells (EpiLC) (Data ref: Khoueiry et al, 2017), are indicated below. RNA-seq TPM expression of *Slc10a4* per group are shown on the right. n = 4 KO matched to n = 4 WT and n = 5 HET in 30 ppm FA; n = 4 KO matched to n = 3 WT and n = 3 HET in 3 ppm FA; n = 6 KO and n = 3 WT and n = 6 HET in 0.1 ppm FA. Error bar represents mean ± SEM.

WT and one KO embryo in excess FA groups) which were extreme outliers. Those two outliers were obtained from embryos that displayed more prominent malformation and decay and henceforth excluded in downstream analysis (Appendix Fig. S7A).

To find differentially expressed genes (DEGs) resulting from loss of *Tet1*, we performed pairwise comparisons between KO vs WT, or HET vs WT, per diet group. Surprisingly, more than 79% of DEGs (false discovery rate (FDR) adjusted *P*-value < 0.05) were in the excess FA group comparing KO vs WT and almost all were downregulated (176 down, 17 up) with the loss of TET1 (Fig. 5A). These downregulated DEGs were enriched in GO terms related to neurotransmission and synaptic function, including "vesicle-mediated transport in synapse", "neurotransmitter secretion", and "synaptic vesicle cycle" (Fig. 5B). Notably, they include a family of genes encoding solute carriers (SLC) facilitating transmembrane transport of L-glutamate and γ-amniobutyric acid (GABA), which are both crucial neurotransmitters in emotion and cognition (Jie et al, 2018; Mohler, 2009) (Fig. 5C). The loss of gene expression related to neurotransmitter and synaptic function aligns with the observed loss of phosphatidylcholine and neurotransmitter metabolites in KO embryos under excess FA (Fig. 3G). Pairwise comparisons between HET vs WT embryos per diet group yielded minimal DEGs, indicating *Tet1* haploinsufficiency has negligible impacts on gene expression in embryonic brains by this developmental stage (Fig. 5A).

KO versus WT brains in the control FA diet group exhibited few DEGs, suggesting that loss of TET1 alone minimally influences gene expression by E11.5, unlike in mouse anterior neuronal progenitor cells (antNPCs) generated by in vitro differentiation from ESCs to mimic the E8.5 neuroectoderm (Data ref: van der Veer et al, 2023). Puzzled by this observation, we examined RNA-seq IGV tracks over the *Tet1* gene locus and observed transcript expression in WT E11.5 brains, elevated from the low levels detected in WT antNPCs and consistent with the post-gastrulation re-activation of *Tet1* expression in the embryonic brain (Data ref: Khoueiry et al, 2017). However, expression of a somatic *Tet1* isoform from a downstream TSS in KO brains was also detectable (Appendix Fig. S7B). We further confirmed the transcript level of

*Tet1* by quantitative RT-PCR (qPCR) analysis using primers targeting the TSS of the embryonic *Tet1* full-length isoform (exon 1a-2) and the first exon of the short isoform (exon 4–5, annotated according to the full-length transcript) (Bartoccetti et al, 2020; Sohni et al, 2015). Indeed, we observed a complete absence of *Tet1* full-length transcript in E11.5 KO brains but detected low expression of the short isoform in both WT and KO at this stage (Appendix Fig. S7C). The short TET1 isoform, which is missing the N-terminus CXXC domain and exhibits reduced global chromatin binding compared to the full-length isoform (Zhang et al, 2016), may be sufficient to rescue gene expression but not DNA hypermethylation at CpG island promoter regions where full-length TET1 predominantly occupies. Moreover, both full-length and short-isoform TET1 proteins were undetectable by Western blots in WT and KO brains at E11.5, indicating that the presence of a short TET1 isoform in the KO samples may not have a significant impact on DNA methylation (Appendix Fig. S7D). Consistent with low protein expression, 5hmC in E11.5 WT and KO brain tissues was detectable by DNA dot blot analysis only at low levels close to background levels relative to strong detection in mouse ESCs, also indicating no compensation for DNA demethylation by TET2 and TET3 (Appendix Fig. S7E). The prevalence of hyper DMRs resulting from loss of *Tet1* under control diet, with minimal DEGs, suggests that elevated DNA methylation has largely silent effects on gene expression at E11.5 or may be persistent aberrations originating from an earlier stage in development.

Collating 206 downregulated DEGs identified from KO vs WT per diet group (Fig. 5A), we interrogated CpG methylation levels at their gene promoters, of which 161 showed sufficient promoter coverage by RRBS. In a heatmap representation (Fig. 5D), a subset (62) of KO downregulated DEGs in all FA diet groups clearly exhibited promoter (TSS ± 2.5 kb) hypermethylation (more than 5% difference compared to the WT), independently of embryo sex and phenotype. Re-analysis of the remaining 144 DEGs revealed that 99 were associated with gene body hypermethylation. Among these DEGs, promoter or gene body hypermethylation were evident in KO brains in all three FA diets, but gene expression was lost in KO only in excess FA diet. In FA depleted diet condition, all genes

were downregulated regardless of *Tet1* genotype. These expression profiles affirm that the 176 downregulated DEGs in *Tet1* KO in excess FA diet are FA-responsive genes, whose induction by FA supplementation requires DNA demethylation by TET1.

Next, to determine the effects of FA diet alone on gene expression, we conducted pairwise comparisons of both excess and depleted FA diet groups with the control, per *Tet1* genotype. Notably, FA deficiency exerts the most pronounced effect on gene expression in HET embryos, resulting in 265 up- and 777 downregulated DEGs (FDR adjusted $P$-value $< 0.05$) (Fig. 5E), highlighting a synergistic impact between *Tet1* haploinsufficiency and folate depletion. The more extensive DEG signature in HET embryos deprived of FA may account for the significant increase in the occurrence of brain deformities (Fig. 2C) compared to HET embryos under control diet. FA deficiency also impacted WT brains with 66 up- and 373 downregulated DEGs but had negligible effects in KO brains (Fig. 5E). The downregulated DEGs resulting from FA depletion in both WT and HET samples were most significantly enriched in GO terms related to neurotransmission and synaptic function, notably "synapse organization", "axonogenesis", and "neurotransmitter transport" (Appendix Fig. S7F,G), strongly resembling DEGs resulting from *Tet1* KO in excess FA diet (Fig. 5B). The predominantly loss of gene expression caused by FA depletion in WT embryos was associated with reduced global histone acetylation, but not with global changes in the facultative heterochromatin mark histone H3 lysine 27 tri-methylation (Appendix Fig. S7H). Global histone deacetylation may be caused by lower gene expression of *Acss2* encoding acetyl-CoA synthetase 2, which catalyzes the conversion of acetate to acetyl-CoA, under FA deficiency, although the differences did not reach statistical significance (Appendix Fig. S7I).

Given that DEGs were observed primarily in three pairwise comparisons—KO vs WT in 30 ppm FA diet, 0.1 ppm vs 3 ppm in WT, and 0.1 ppm vs 3 ppm HET embryos—we overlapped this composite set of DEGs to identify a subset common to all, which we considered to be genes whose expression are responsive to dietary FA fluctuations. From the Venn overlap, we identified 71 common DEGs (Fig. 5F). GO analysis of these 71 genes showed a robust enrichment of terms related to neurotransmission and synaptic function, including "neurotransmitter transport", "vesicle-mediated transport in synapse", and "synapse organization" in the top three (Fig. 5G). Analysis of absolute expression levels confirmed that these 71 genes failed to be upregulated selectively in KO embryos exposed to excess FA, compared to WT and HET embryos (Fig. 5H). In this scenario, an excess of FA in the absence of TET1 may paradoxically mimic FA deficiency by a downregulation of multiple solute carriers involved in neurotransmitter transport, resembling a compensatory response to reduce influx of FA. As depicted above in the association between promoter/gene body methylation and gene expression, while hyper DMRs resulting from *Tet1* KO were prevalent across all diet groups, for instance at the genes encoding solute carriers *Slc10a4* and *Slc32a1*, these hyper DMRs were associated with a significant loss of gene expression only under excess FA (Fig. 5I; Appendix Fig. S7J), suggesting that a synergy between excess FA status and DNA hypermethylation leads to silencing of these genes.

## Regulation of the proton-coupled folate transporter gene by TET1 is FA-sensitive

Folate transport across cell membranes involves three distinct mechanisms, via receptor-mediated endocytosis of the folate

receptor alpha and beta (FRα, gene *Folr1*; FRβ, gene *Folr2*), via the reduced folate carrier (RFC, gene *Slc19a1*) in systemic tissues, and via the proton-coupled folate transporter (PCFT, gene *Slc46a1*) located primarily in the kidney, gastrointestinal tract, and the choroid plexus in the ventricles of the brain (Sangha et al, 2022). The *Slc46a1* gene has a CpG island promoter known to be regulated via DNA methylation (Gonen et al, 2008). Normalized transcripts per million (TPM) values of *Slc46a1, Slc19a1, Folr1*, and *Folr2* in our RNA-seq analysis of 129S6.Cg E11.5 brains did not show any significant changes in expression of these genes as a result of altering *Tet1* genotype or FA diet (Appendix Fig. S8A). In CD1 E11.5 embryos, we observed only a modest downregulation of *Slc46a1* in the KO brains under both excess and normal FA diet conditions (Appendix Fig. S8B). Nonetheless, our previous WGBS revealed CpG hypermethylation at the *Slc19a1* (increase of ~3%) and *Slc46a1* (~13%) promoter regions (TSS ± 1 kb) in KO antNPCs relative to WT, while the promoter regions of *Folr1* and *Flor2* both lack CpG islands and detectable DMRs (van der Veer et al, 2023; Data ref: van der Veer et al, 2023) (Fig. 6A; Appendix Fig. S8C). Moreover, TET1 occupies the promoter regions at *Slc19a1* and *Slc46a1*, but not at *Folr1* and *Folr2*, in primed epiblast-like cells (EpiLCs), where stronger binding was detected at *Slc46a1* than at *Slc19a1* (Khoueiry et al, 2017; Data ref: Khoueiry et al, 2017) (Fig. 6B; Appendix Fig. S8C). Therefore, *Slc46a1* appears to be the most likely TET1 target among known folate carriers, prompting us to validate its potential regulation by TET1 in post-gastrulation neuroectoderm cells prior to neural tube closure.

To obtain an in vitro correlate of E8.5 rostral neural cells under standard defined culture media containing high FA supplements (>5 mg/L), we performed in vitro differentiation of WT and TET1 KO ESCs, derived from the mouse blastocysts from 129S6.B6-*Tet1*[+/−] x *Tet1*[+/−] intercrosses, into antNPCs (van der Veer et al, 2024). QPCR showed a trend of reduced *Slc46a1* transcript levels in KO compared to WT antNPCs that reached borderline significance ($P \leq 0.07$), supported by a Western blot analysis that revealed notable loss in PCFT protein expression in *Tet1* KO cells (Fig. 6C,D). These results suggest that PCFT expression in neural tissues in early development may be moderately dependent on TET1.

Finally, we asked whether dietary FA levels affect *Slc46a1* promoter methylation levels in E11.5 KO embryos, reasoning that any DNA methylation changes induced by diet x genotype effects would persist until E11.5. Because the WGBS DMR at *Slc46a1* identified in KO versus WT antNPCs was not covered by RRBS, we conducted targeted bisulfite sequencing over the region surrounding the *Slc46a1* promoter CpG island to re-analyze whole embryos collected from dams on modified FA diets on the 129S6.Cg background (Appendix Fig. S8D,E). Interestingly, FA excess drove a significant elevation in CpG methylation at the *Slc46a1* CpGi shore in KO embryos, whereas methylation differences were not discernible between WT and KO under control and depleted FA diets (Fig. 6E). These results affirm that dietary folate status can be coupled with *Tet1* function to regulate promoter DNA methylation of multiple membrane transporters, including a folate transporter.

## Discussion

In this study, we determined how the DNA dioxygenase TET1, a newly identified candidate gene for NTDs, connects FA one-carbon

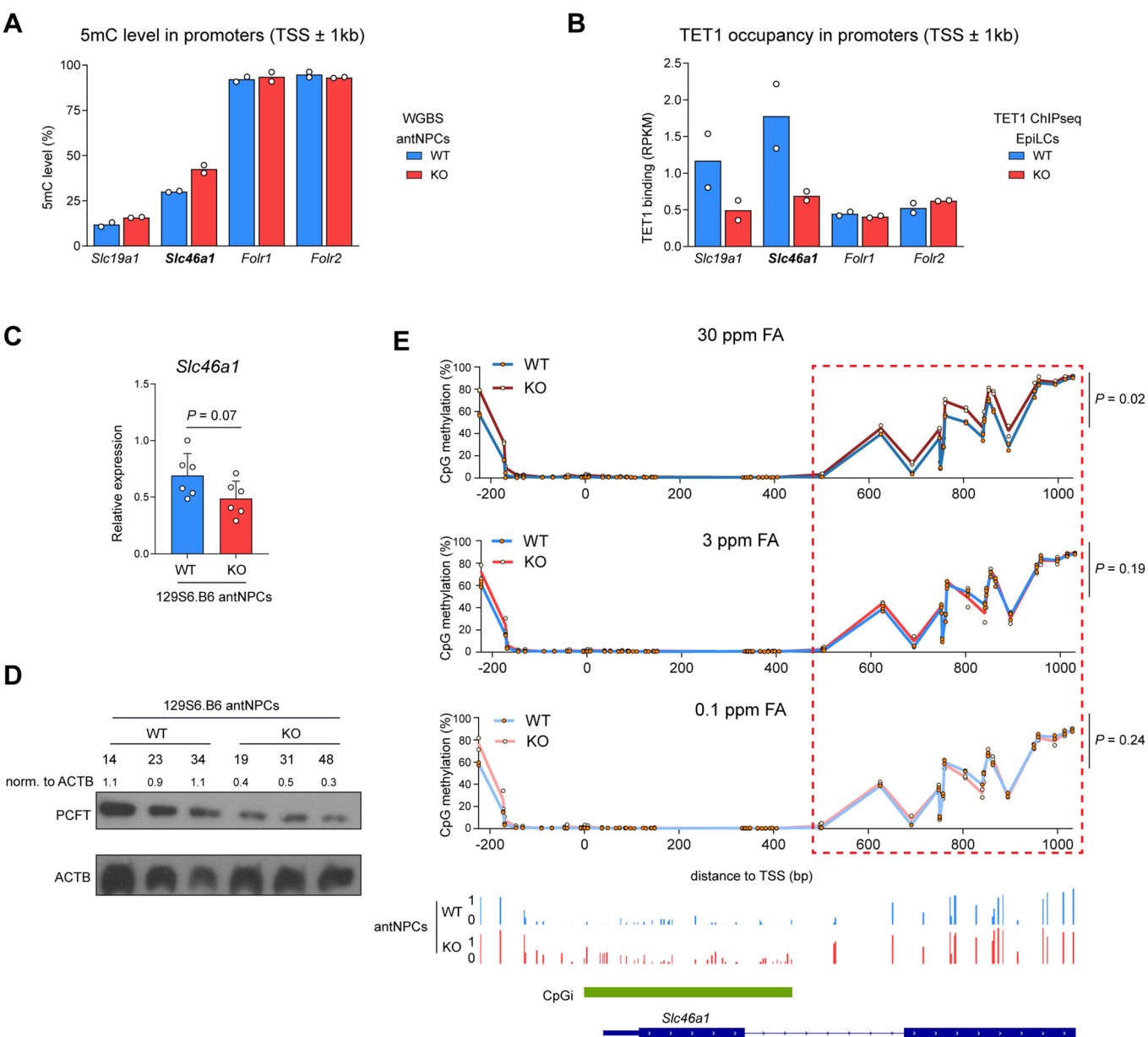

**Figure 6. DNA hypermethylation on the promoter region of folate transporter gene *Slc46a1* as a result of excess FA coupled to Tet1 loss.**

(A) 5mC levels identified by WGBS on the promoter regions (±1 kb of TSS) of *Slc19a1*, *Slc46a1*, *Folr1*, and *Folr2* in WT and KO antNPCs of a B6129S6F1 strain (day 5 of in vitro differentiation under Wnt inhibition). (Data ref: van der Veer et al, 2023). (B) TET1 binding shown by TET1 chromatin immunoprecipitation-sequencing (ChIPseq) on the promoter regions (±1 kb of TSS) of *Slc19a1*, *Slc46a1*, *Folr1* and *Folr2* in WT and KO EpiLCs (Data ref: Khoueiry et al, 2017). (C) QPCR analysis of *Slc46a1* in KO and WT antNPCs of a 129S6.B6 incipient congenic strain. Data are shown as mean ± SEM of n = 6 biological replicates from two independent differentiations using three different ESC lines per genotype. P value is calculated by t-test. (D) Western blot of PCFT in 129S6.B6 KO and WT antNPCs. Numbers below WT and KO antNPCs indicate clonal identities. Densitometric band intensities of PCFT normalized to ACTB (beta-actin loading control) are shown above each lane of the blot. (E) Targeted amplicon bisulfite sequencing analysis of CpG methylation levels at the promoter region of *Slc46a1* in WT and KO whole embryos exposed to modified maternal FA diet. n = 2 individual embryos for each genotype per diet. X-axis 0 indicates position of the TSS. IGV tracks below show CpG methylation levels in F1 strain WT and KO antNPCs over the corresponding region (Data ref: van der Veer et al, 2023). P values are calculated by paired 2-way ANOVA, based on base position (CpG) as a paired factor and genotype as an independent factor, indicating the statistical difference caused by the genotype factor. Source data are available online for this figure.

metabolism, DNA methylation and gene expression to influence embryonic development post-NTC. We uncovered a synergy between excess FA status and loss of *Tet1*, which results in the downregulation of membrane transporters and disruption of FA intake. On the other hand, the effects of FA deficiency can be exacerbated by *Tet1* haploinsufficiency, impacting cellular methylation potential and NTD susceptibility. Overall, the interplay between epigenetic regulation by TET1 and folate 1C metabolism converges on the regulation of gene expression of multiple solute carriers and steady-state levels of phospholipid and neurotransmitter metabolites, ultimately affecting neurodevelopment beyond NTC.

The most recent recommendation by the US Preventive Services Task Force re-affirmed the substantial NTD-protective benefit of FA supplementation, when taken at 0.4 to 0.8 mg daily during the peri-conceptual period (Viswanathan et al, 2023). However, excessive maternal FA intake can be as high as 5 mg/day in certain medical conditions such as those linked to low folate levels (Paniz et al, 2017; Paniz et al, 2019). Whether this may introduce aberrant DNA methylation events or de novo point mutations that predispose the offspring to chronic diseases in adulthood is a concern that warrants further investigation (Cao et al, 2023; Fardous and Heydari, 2023; Murray et al, 2018). The influence of excessive maternal dietary folate on DNA methylation has been reported to affect specific loci including imprinted genes (Barua et al, 2016; Haggarty et al, 2013; Ly et al, 2020; Tserga et al, 2017). In this study, we used 30 ppm FA as a dose representing a 10-fold surplus of FA in rodent diet, in accordance with other studies (Burren et al, 2008; Cao et al, 2023; Harlan De Crescenzo et al, 2021). This high-dose FA supplementation may cause teratogenic effects on fetal brain development and confer susceptibility to embryonic defects (Asadi-Pooya, 2015; Pickell et al, 2011; Wang and Chen, 2016). Indeed, we observed mild teratogenic effects of excess FA inducing brain malformation during embryonic development.

Nonetheless, our genome-wide profiling uncovered hundreds of DMRs predominantly with gains of DNA methylation in WT mouse embryos, which were selected for analysis based on mostly normal morphology. The impact of excess maternal FA on the WT DNA methylome is generally not associated with gene expression changes by E11.5 but can result in long-lasting silencing of gene pathways strongly enriched for neurotransmitter transport in the absence of TET1. Our WGBS analysis affirmed that loss of TET1 resulted in a marginal increase in DNA hypermethylated loci induced by excessive FA, shifting the hyper DMRs from promoter proximal to more distal regions. This finding aligns with previous WGBS studies by us and others localizing TET1 activities to CpGi flanking regions (shores), where DNA demethylation serves to counter DNMT activities to preserve CpGi promoters in a fully unmethylated state, as well as distal enhancers (Dixon et al, 2021; Liao et al, 2015; Sun et al, 2019; Data ref: van der Veer et al, 2023; Wu and Zhang, 2017). Using targeted bisulfite sequencing at Slc46a1, we confirmed that excess FA can indeed elevate DNA methylation levels in KO relative to WT embryos at a CpGi shore region not covered by RRBS. The compromised FA intake despite excess FA supplementation in Tet1 KO embryos, which could result from downregulation of Slc46a1, is likely diverting a more limiting pool of one-carbon units towards methylation of CpGi shore and distal regions in the absence of TET1.

In comparison with excessive FA supplementation, FA deprivation induced DMRs more extensively, almost equally in both directions of gain and loss in DNA methylation in WT embryos. We characterized subtle differences in genomic localization between these FA deficiency-induced hyper and hypo DMRs but clarified that the net effect of both converges on common pathways associated with development and Wnt signaling, concomitantly with gene expression changes predominantly downwards. While the limited coverage of RRBS did not provide a comprehensive view of how FA deficiency affects DNA methylation patterns over the whole genome, we showed that losing one copy of Tet1 allele in HET embryos is sufficient to alter DNA methylation changes caused by deficient FA diet (Fig. 4A, B), supporting an elevated

sensitivity of DNA methylation status to Tet1 gene dosage under conditions of FA deficiency. Coincidentally, we observed the highest penetrance of cranial malformations under combined deficiency in FA and Tet1.

Although excess and depleted maternal FA status showed clearly distinct impacts on steady-state levels of FA, SAM/SAH and serine/glycine ratios, we also observed several instances in which excessive FA in the absence of TET1 mimics a state of pseudo-FA deficiency: (1) reduction in RRBS promoter-associated hyper DMRs in Tet1 KO; (2) downregulation of multiple membrane transporters and gene pathways associated with neurotransmitter transport and function and (3) loss of phospholipid metabolites, especially phosphatidylcholines (PCs). Of note, PCs are the most abundant phospholipids in mammalian cellular membranes, and in general their synthesis requires methylation reactions involving phosphatidylethanolamine methyltransferase (PEMT) and betaine-homocysteine methyltransferase (BHMT), both dependent on folate cycling of one-carbon via SAM (Li and Vance, 2008; Stead et al, 2006; Ueland, 2011). The absence of Tet1 may shift SAM utilization under excess folate cycling towards uncurbed DNMT activities at the expense of PC synthesis, ultimately affecting transport of neurotransmitters and synapse formation. In this regard, the loss of Tet1 mimics the reduction of MTHFR protein expression and activity by excess FA, which creates a pseudo-MTHFR deficiency that results in lower SAM/SAH ratio and alterations on lipid metabolism (Christensen et al, 2015). These results underscore the need to evaluate high-dose maternal FA supplementation in the context of potential nutritional stressors that may disrupt cellular dioxygenase functions, particularly for adverse impact on fetal neurogenesis and post-natal neurological functions.

Despite folate bioavailability being a major nutritional modifier of NTD risk, mutations directly affecting folate-related genes are relatively rare in cases of mouse NTDs (Harris and Juriloff, 2010; Wallingford et al, 2013). Of these, only a few, such as Mthfd1l (Momb et al, 2013), Gldc (Pai et al, 2015), and Slc25a32 (Kim et al, 2018), result in NTDs under folate deprivation, and all are resistant to FA supplementation. Interestingly, supplementation with formate, a THF-independent donor, circumvents the metabolic blocks caused by these mutations and rescues the FA resistance (Kim et al, 2018; Pai et al, 2015). While over 300 mouse mutant models for NTDs exist in the literature, only a fraction have been characterized for responsiveness to FA supplementation (Gray and Ross, 2009; Marean et al, 2011). Our detailed study of Tet1 mutants suggests that epigenetic dysfunction resulting from TET1 deficiency may be a basis for FA resistance across strains. Possibly, Tet1$^{-/-}$ embryos are less able to absorb FA due to downregulation of Slc46a1/PCFT at the onset of neurulation (in 129S6.Cg and CD1 backgrounds). On the other hand, FA deprivation exposes an effect of Tet1 haploinsufficiency on methylation potential and increased occurrence of congenital structural malformations including NTD. These observations suggest that the risk for birth defects associated with FA status may be modified by nutritional and other environmental factors that affect TET activity and function, such as vitamin C (Blaschke et al, 2013). By this rationale, FA supplementation in combination with vitamin C may offer further protective benefits against birth defects. By reversing aberrant DNA methylation by informed nutritional intervention, FA-resistant cases of NTDs may potentially be converted into FA-responsive ones.

In the mouse, NTC begins at E8.5, resulting in closure of the anterior and midbrain-hindbrain neuropores by E9.5 and the posterior neuropore by E10.5 (Yamaguchi and Miura, 2013). The experimental endpoint in this study at E11.5 would not have captured fully the dynamics of metabolomic and transcriptomic changes due to diet and genotype interactions during NTC. Indeed, our transcriptome analysis of embryonic brain tissues hardly detected any DEGs caused by loss of TET1 in the control diet but revealed persistent downregulation of solute transporters as a result of excess FA treatment combined with the loss of TET1. Yet, the genome-wide DNA methylome analysis adequately captured widespread DMRs that more likely accumulated from diet x genotype manipulations during NTC, in line with a long-lasting presence of DNA methylation changes. For practical reasons, this study of E11.5 stage embryos enabled us to access a broader range of structural birth defects in addition to NTDs and also accurately dissect brain tissues (forebrain, midbrain, and hindbrain) to obtain tissue-specific epigenome and transcriptome profiles. Insights from this study will be the basis for further investigations of *Tet1* genotype x FA diet interactions earlier during the initiation of NTC, and later during postnatal development.

In summary, our study unravels one intricate layer of gene-environmental interactions influencing embryonic neurodevelopment post-NTC, involving TET1 and folate one-carbon metabolism. By decoding the interplay of TET1 with the genome, metabolome, and epigenome, our findings shed light on FA-sensitive genomic loci and uncover a potential epigenetic mechanism regulating FA responsiveness. Targeted intervention strategies tailored to rectify these pathways with individualized micronutrient supplementation plans may have a major impact in preventing FA-resistant NTDs, a major focus of public health concern in the era post-FA fortification.

# Methods

**Reagents and tools table**

| Reagent/Resource | Reference or Source | Identifier or Catalog Number |
|---|---|---|
| **Experimental Models** | | |
| C57BL/6J | Jackson Lab | JAX strain #000664 |
| B6-Tet1<tm1Koh> mouse | Khoueiry et al, 2017 | Mouse Genome Informatics (MGI) ID: 6099090 |
| CD-1 | Jackson Lab | JAX stock #034608 |
| 129S6/SvEvTac inbred mouse strain | Taconic Biosciences | MGI ID: 3044417 |
| **Antibodies** | | |
| Anti-Histone H3ac | Active-Motif | 39139 |
| Anti-H3 | Abcam | ab1791 |
| Anti-H3K27me3 | Upstate | 07-449 |
| Anti-PCFT | Abcam | ab25134 |
| Anti-TET1 | Genetex | GTX125888 |
| Anti-ACTB | Cell-signaling | 4970 |
| Anti-mouse | DAKO | P0447 |

| Reagent/Resource | Reference or Source | Identifier or Catalog Number |
|---|---|---|
| Anti-rabbit | DAKO | P0217 |
| Anti-5hmC | Active Motif | 39769 |
| Anti-dsDNA | Abcam | ab27156 |
| **Oligonucleotides and other sequence-based reagents** | | |
| PCR primers | This study | Table 1 |
| qPCR primers | This study; Sohni et al, 2015 | Table 2 |
| Bisulfite PCR primers | This study | Table 3 |
| **Chemicals, Enzymes and other reagents** | | |
| phosphatase inhibitor cocktail 2 and 3 | Sigma-Aldrich | P5726 and P0044 |
| protease inhibitor cocktail | Roche | 11836153001 |
| Clarity Western ECL substrate | Bio-Rad | 1705060 |
| PureGene Genomic DNA extraction kit | Invitrogen | K182001 |
| Amersham Hybond positive charged nylon membrane | Cytiva | RPN119B |
| 5-dhCMP | Jena Bioscience | NU-932S |
| RNeasy plus mini kit | Qiagen | 74136 |
| Superscript III cDNA synthesis | Thermo Fisher | 11752-050 |
| SYBR-green PCR master mix | Thermo Fisher | 11733-046 |
| Ovation RRBS Methyl-Seq kit | Tecan/Nugen | 0553-32 |
| TaqI | NEB | R0149S |
| 96-Plex Adaptor Plate | Tecan | S02223 |
| Agencourt AMPure XP beads | Beckman Coulter | A63881 |
| Qubit™ dsDNA HS Assay kit | Invitrogen | Q32854 |
| High Sensitivity DNA analysis kit | Agilent | 5067-4626 |
| RNA 600 Nano kit | Agilent | 5067-1511 |
| KAPA Stranded mRNA Hyperprep Kit (96rxn) | Roche | KK8581 |
| KAPA-single index adapters | Roche | KK8700d |
| EpiTect Fast DNA Bisulfite kit | Qiagen | 59824 |
| PlatinumTM Taq DNA polymerase High Fidelity | Invitrogen | 11304-011 |
| **Software** | | |
| QuantStudio Real-Time PCR software (v1.3) | Applied Biosystems | |
| MassHunter Profinder 8.0 and MassProfiler Professional (MPP) 15.1 software | Agilent technologies | |
| GraphPad Prism 9 | GraphPad Software Inc. | |
| **Other** | | |
| NovaSeq S4 | Illumina | |
| Illumina Hiseq4000 | Illumina | |
| Illumina NovaSeq 6000 | Illumina | |

**Table 1. PCR primers.**

| Type | Oligo name | Sequence (5′ to 3′) |
|---|---|---|
| *Tet1<tm1>* genotyping | Tet1tm1-Gtype-7F | TTGGCAACACCTCCAGATT |
| | Tet1tm1-Gtype-6R | CGGATTGACCGTAATGGGATAG |
| | Tet1tm1-Gtype-10R | GCTTTGATGTCTTCGTCTTCATC |
| Sex genotyping | SX-F | GATGATTTGAGTGGAAATGTGAGGTA |
| | SX-R | CTTATGTTTATAGGCATGCACCATGTA |

## Animal maintenance, mouse strain breeding, and timed pregnancies

All experimental procedures on mice were reviewed and approved under project P101/2016 by the KU Leuven Ethical Committee for Animal Experimentation in compliance with the European Directive 2010/63/EU. *Tet1*tm1Koh mice were maintained on a 12-h day/night cycle with grain-based 'chow' maintenance complete feed (Ssniff, Germany) used at the animal facility at KU Leuven. Breeding pairs were fed with Ssniff breeding complete feed and set up by crossing heterozygous with wild-type mice to obtain heterozygous offspring for timed pregnancy experiments. Both regular maintenance and breeding diet contained 7 ppm FA. Custom diet containing modified FA (TestDiet, US) was based on a modification of AIN-93G semi-purified diet (57W5) with no added folate and 1% succinylsulfathiazole. Unless stated otherwise, mice in experimental timed mating were fed Ssniff maintenance complete feed (regular chow). Pups were genotyped after weaning by performing PCR using primers as listed in Table 1.

The *Tet1*tm1Koh line was originally created as a C57Bl/6 (B6) congenic strain (Khoueiry et al, 2017) and henceforth maintained by regular backcrossing to C57Bl/6J obtained from Jackson Lab. To transfer the *Tet1*tm1Koh allele to alternative genetic backgrounds, we outcrossed it (>3 generations) to the outbred CD1 stock. Subsequently, we backcrossed B6-*Tet1*tm1Koh mice to an inbred 129S6 (also known as 129SvEvTac) strain. We considered mice obtained from 5 to 7 generations of backcrossing as incipient congenic, and those after >10 generations as congenic. Wherever indicated in the text, the number of backcross generations is prefixed by the letter N.

To obtain embryos from timed pregnancies, *Tet1*+/− mice were paired and monitored for 5 days, checking every morning for the presence of a copulation plug. The morning of detecting a copulation plug was considered to be E0.5. Pregnant dams were sacrificed at indicated time-points using cervical dislocation. Embryos were dissected from individual decidua in ice-cold PBS. Following excision of the yolk sac, embryos were imaged using Leica Application Suite V4, and assigned a preliminary score according to the EMAP definition by a researcher. These scores were subsequently validated by two or three independent researchers based on images. Exencephaly was scored as "NTD". For FA-modulation experiments, we also observed other malformations, that were classified as brain malformation, gross malformation of multiple visceral organs, and growth delay based on somite counts. Whole embryos were collected and snap frozen

in liquid nitrogen or fixed in 4% PFA overnight at 4 °C. Embryonic brains were dissected if needed and snap frozen in liquid nitrogen. For embryos ≥E10.5, a portion of the yolk sac, free of maternal blood, was collected for genotyping.

## Modulation of maternal FA status

To perform intraperitoneal (IP) injections of FA, CD1-*Tet1*tm1Koh *Tet1*+/− females were adapted post-weaning for 4 weeks to a 2.1 ppm FA diet. After the detection of a copulation plug during mating, females were IP injected daily from E5.5 with 10 mg/kg FA until the experimental endpoint. Similarly, to administer methionine, *Tet1*+/− females were fed regular chow and IP injected with 70 mg/kg methionine from E5.5 daily until the experimental endpoint.

To modify maternal FA status using custom diets, *Tet1*+/− females were adapted post-weaning for 4 weeks to the modified FA diets in groups of up to 5 per cage. As much as possible, littermate females were randomly assigned to different diet groups. Stud *Tet1*+/− males were re-used in timed pregnancies on a rotational basis across all diets to randomize their exposure to custom diets during the periods of timed mating (up to 7 days) and otherwise maintained on regular chow diet. To determine whether the addition of SST in our custom diets by itself affected NTD phenotypic penetrance, additional groups of CD1-*Tet1*tm1Koh mice were adapted with customized diets containing 2.7 or 7 ppm FA (matching level in regular chow) supplemented with or without 1% SST. At experimental endpoints, dams were euthanized for embryo collection. The sex and genotype of individual E11.5 embryos were determined by PCR analysis with isolated yolk sac tissues. For the collection of biological replicates in next-generation sequencing analyses, we selected a *Tet1*−/− embryo matched by somite numbers to a littermate WT and/or *Tet1*+/− embryo of the same sex. For each downstream assay, we collected 4 to 6 replicates from independent litters, representative of the phenotypes observed and matching numbers of male and female replicates.

## Western blotting

Cells were washed with ice-cold PBS and lysed on plate in ice-cold RIPA buffer (50 mM Tris at pH 8.0, 150 mM NaCl, 0.2 mM EDTA, 1% NP-40, 0.5% sodium deoxycholate, 0.1% SDS, containing 1 mM phenylmethylsulfonyl fluoride, 0.5 mM DTT, phosphatase inhibitor cocktail 2 and 3 (Sigma-Aldrich, P5726 and P0044) and protease inhibitor cocktail (Roche, 11836153001)). Lysates were scraped off the plates and transferred to pre-cooled microcentrifuge tubes, incubated on ice for 30 minutes, subsequently passed through a 26-gauge needle and clarified by centrifugation for 15 min at 16,000 r.c.f. at 4 °C, after which supernatant was collected and stored at −80 °C. Protein concentration was measured using Bradford assay in a 96-well microplate format. Whole cell lysate samples were prepared in 1x Laemmli sample buffer (62.5 mM Tris-HCl at pH 6.8, 2.5% SDS, 0.002% bromophenol blue, 5% β-mercaptoethanol, 10% glycerol) and boiled for 10 min at 95 °C. 10 to 15 μg of protein was loaded on an 8% (for TET proteins, >200 kD) or 10% (other proteins) SDS–polyacrylamide gel, electrophoresed in 1X running buffer (25 mM Tris, 192 mM glycine, 0.1% SDS) and then transferred to a PVDF membrane with transfer buffer (25 mM Tris, 192 mM glycine and 20% methanol; 0.1% SDS added when

**Table 2.** qPCR primers.

| | Oligo name | Sequence (5' to 3') |
|---|---|---|
| *Slc46a1* | Slc46a1_qPCR_FW | GCTCACCACACAGTACCTTT |
| | Slc46a1_qPCR_RV | ATGTAGAGGGTCCAGTGAGAT |
| *Tet1* Exon1a-2 | Tet1_qPCR_exon1a_FW | CTGCCTCTTCTACGGGAACATTCG |
| | Tet1_qPCR_exon 2_RV | GCCTGCTTTGATGTCTTCGTCTTC |
| *Tet1* Exon4-5 | Tet1_qPCR_exon4_FW | GAGGGAAAAGAAGCCCAAAG |
| | Tet1_qPCR_exon5_RV | CGCCTGCATTCTTCCTTACA |

transferring TET proteins). Membranes were blocked using 5% skim milk in Tris-buffered saline containing 0.1% Tween-20 (TBS-T) prior to incubation with primary antibodies diluted in the blocking buffer for overnight at 4 °C, and subsequently with HRP-conjugated secondary antibodies for 1 h at room temperature. The signal was detected using Clarity Western ECL substrate (Bio-Rad 1705060) and developed on a light-sensitive film using a AGFA Curix 60 Film Processor. Primary antibodies (listed with catalog number and dilution factor) used in this study were: anti-Histone H3ac (Active-Motif, 39139, 1:1000), anti-H3 (Abcam, ab1791, 1:5000), anti-H3K27me3 (Upstate, 07-449, 1:1000), anti-PCFT (Abcam, ab25134, 1:1000), anti-TET1 (Genetex, GTX125888, 1:1000), anti-ACTB (Cell-signaling, 4970, 1:4000). Secondary antibodies were: anti-mouse (DAKO, P0447, 1:5000), anti-rabbit (DAKO, P0217, 1:5000).

## Dot blotting

Dot blotting was performed as previously described (van der Veer et al, 2023). Briefly, genomic DNA (gDNA) was extracted using a PureGene Genomic DNA extraction kit (Invitrogen, K182001) according to manufacturer's instructions. 250 ng of gDNA was serially diluted two-fold in nuclease-free water followed by denaturation in 0.4 M NaOH/10 mM EDTA at 95 °C for 10 min, then neutralized in an equal volume of ice-cold 2 M ammonium acetate and kept on ice for 10 min. An Amersham Hybond positive charged nylon membrane (Cytiva, RPN119B) were prewetted with $H_2O$ for 20 min, and followed by 6X SSC for 20 min. Samples were then spotted onto the prewet membrane with a 96-well Bio-Rad Bio-Dot apparatus. The spotted membrane was subsequently washed excessively with 2X SSC, air-dried for 5–10 min and UV cross-linked two times at 120,000 $\mu J/cm^2$ using a UVP HL-2000 HybriLinker. The membrane was incubated with antibodies as described for Western blot. Primary antibodies used were: anti-5hmC (Active Motif, 39769; 1:1000) and anti-dsDNA (Abcam, ab27156, 1:5000; which cross-reacts with both single and double-stranded DNA). On the membrane, a serial dilution of 0.1 ng of a PCR product generated using 5-dhCMP (Jena Bioscience, NU-932S) instead of 5-dCMP in the dNTP mix was spotted as a positive control.

## Quantitative reverse transcription-polymerase chain reaction (qPCR)

Total RNA was extracted using a RNeasy plus mini kit (Qiagen, 74136) according to the manufacturer's instructions. 200 ng–1 µg of

RNA was used for reverse transcription (RT) reactions using Superscript III cDNA synthesis (Thermo Fisher, 11752-050), according to the manufacturer's instructions. Quantitative real-time PCR reactions were set up in technical triplicates using SYBR-green PCR master mix (Thermo Fisher, 11733-046) supplemented with 5 µM primers (Table 2) and 1:100 of each cDNA reaction on a 384-well ViiA7 real-time PCR system (Applied Biosystems). Expression levels of target genes was calculated according to the 2–ΔΔCt method using QuantStudio Real-Time PCR software (v1.3), for which *Gapdh* expression was used for normalization and fold induction was calculated relative to expression in control ESCs.

## Mass spectrometry

E11.5 whole embryos were snap frozen for storage at −80 °C and subsequently processed for high performance liquid chromatography coupled to tandem mass spectrometry (HPLC-MS/MS) analysis as previously described (Leung et al, 2017). Polar metabolites were extracted in −70 °C 80% methanol/water for untargeted metabolomic analysis using a platform comprised of an Agilent Model 1290 Infinity II liquid chromatography system coupled to an Agilent 6550 iFunnel time-of-flight MS analyzer. Comprehensive chromatography of metabolites utilized a combination of aqueous normal phase (ANP) chromatography on a 2.1 mm Diamond Hydride column, and reverse-phase C18 hydrophobic chromatography, operated on both positive- and negative-ion modes for MS-based ratio-metric quantification of metabolites. Mobile phases for aqueous chromatography consisted of (A) 50% isopropanol, containing 0.025% acetic acid, and (B) 90% acetonitrile containing 5 mM ammonium acetate. To eliminate the interference of metal ions on chromatographic peak integrity and electrospray ionization, EDTA is added to the mobile phase at a final concentration of 5 µM and the following gradient applied for separation: 0–1.0 min, 99% B; 1.0–15.00 min, to 20% B; 15.0 to 29.0, 0% B; 29.1 to 37 min, 99% B. Raw data were analyzed using MassHunter Profinder 8.0 and MassProfiler Professional (MPP) 15.1 software (Agilent technologies). Our in-house compound structural identity database included over 938 metabolites in our in-house MassHunter PCDL manager 8.0 database (Agilent) comprised of molecules that span 1C metabolism, vitamin co-factor, amino acid, nucleotides, carbohydrates, lipids, and energy pathways. A molecular formula generator (MFG) algorithm in MPP is used to generate and score empirical molecular formulae, based on a weighted consideration of monoisotopic mass accuracy, isotope abundance ratios, and spacing between isotope peaks. A tentative compound ID is assigned when PCDL database and MFG scores concurred for a given candidate molecule. Metabolite structures are assigned based on monoisotopic neutral masses (<5 ppm mass accuracy), and chromatographic retention times, with confirmation by MS/MS fragmentation pattern matching to reference standards (Keuls et al, 2020; Steele et al, 2022).

The untargeted metabolic profiling approach precluded accurate measurement of bioactive forms of FA, such as 5-methyltetrahydrofolate (5-MTHF) and tetrahydrofolate (THF). As a result, metabolites of these bioactive forms of FA were excluded from the metabolomic data. Non-mammalian metabolites, metabolites with a retention time <1.3 min and/or no confidence on the ID assignment were also excluded. Chiral structure specifications were removed, and

**Table 3.   bisulfite PCR primers.**

|  | Oligo name | Sequence (5′ to 3′) with Adapters |
|---|---|---|
| Slc46a1 | Slc46a1_primer1_FW | CTTTCCCTACACGACGCTCTTCCGATCTAGTAGGGATTAGAGTAGTTGTGAATA |
|  | Slc46a1_primer1_RV | CTGGAGTTCAGACGTGTGCTCTTCCGATCTAACCTTACTTTAAAACCCTAAACCTCT |
|  | Slc46a1_primer2_FW | CTTTCCCTACACGACGCTCTTCCGATCTGAATATTTTAAGAGGTTTAGGGTT |
|  | Slc46a1_primer2_RV | CTGGAGTTCAGACGTGTGCTCTTCCGATCTACTACACCACAAAAATAAACACCA |
|  | Slc46a1_primer3_FW | CTTTCCCTACACGACGCTCTTCCGATCTGGTAGAGTATAAGTATTTTAGTGTGG |
|  | Slc46a1_primer3_RV | CTGGAGTTCAGACGTGTGCTCTTCCGATCTAACTTTATTCTAAAACCAAAAATTCTAAA |
|  | Slc46a1_primer4_FW | CTTTCCCTACACGACGCTCTTCCGATCTAGAATTTTTGGTTTTAGAATAAAG |
|  | Slc46a1_primer4_RV | CTGGAGTTCAGACGTGTGCTCTTCCGATCTCCAAACAATCCCTATAATATATCCTC |
|  | Slc46a1_primer5_FW | CTTTCCCTACACGACGCTCTTCCGATCTGTTTGAAGTGAGGATATATTATAGGGA |
|  | Slc46a1_primer5_RV | CTGGAGTTCAGACGTGTGCTCTTCCGATCTAAACCTAAACCAATCTATACTATCTA |
|  | Slc46a1_primer6_FW | CTTTCCCTACACGACGCTCTTCCGATCTGTAGATAGTATAGATTGGTTTAGGT |
|  | Slc46a1_primer6_RV | CTGGAGTTCAGACGTGTGCTCTTCCGATCTATTAACCAAAAAAACTAAAAACTCCA |
|  | Slc46a1_primer7_FW | CTTTCCCTACACGACGCTCTTCCGATCTATTTTTGTGGTGTAGTTAGAATTG |
|  | Slc46a1_primer7_RV | CTGGAGTTCAGACGTGTGCTCTTCCGATCTCAACCAAAAAAAAATTAACATAACCCTA |
|  | Slc46a1_primer8_FW | CTTTCCCTACACGACGCTCTTCCGATCTGGAGTTTTTAGTTTTTTTGG |
|  | Slc46a1_primer8_RV | CTGGAGTTCAGACGTGTGCTCTTCCGATCTCACAACTACTCTAATCCCTACTACCTTC |

lipid names were standardized according to the Lipidomics Standard Initiative. Statistical analysis was performed first using one-way ANOVA to identify metabolites with significant changes across all groups, and further using 2-way ANOVA to identify significant interactions between the diet and genotype factors ($P < 0.05$). Significant metabolite values were scaled using Z-score and visualized using the pheatmap R package (v. 1.0.12) (Kolde et al, 2012).

## Reduced representation bisulfite sequencing (RRBS) library preparation

Genomic DNA (gDNA) was extracted using the Purelink Genomic DNA mini kit (Invitrogen, K182001) from individual E11.5 brains dissected from the most rostral aspect of the forebrain to the caudal aspect of the hindbrain above the otic vesicle. DNA quality was assessed with NanoDrop One (Thermo Fisher), and absence of RNA contamination was confirmed with a 0.8% agarose gel stained with SYBR Safe. RRBS libraries were prepared to include oxidative bisulfite (oxBS) conversion using the Ovation RRBS Methyl-Seq kit (Tecan/Nugen, 0553-32), according to the manufacturer's instructions. Per sample, 100 ng of gDNA as input was processed through the oxBS workflow to map true 5mC. To increase the coverage, we performed double digestion using MspI (Supplied with the Nugen Kit) and TaqI (NEB, R0149S) in each sample, first with MspI digestion at 37 °C for 60 min, followed by TaqI digestion at 65 °C for another 60 min. After digestion, each library was ligated to a custom single-indexed sequencing adapter (8 bp barcodes from the 96-Plex Adaptor Plate, Tecan, S02223) followed by final repair. Before oxidation and bisulfite conversion, libraries were purified and washed 3× with 80% acetonitrile to get rid of any residual ethanol, followed by incubation at 37 °C for 5 min with denaturing buffer provided with the kit. Libraries were oxidized by the addition of TrueMethyl oxidant solution incubating at 40 °C for 10 min. Immediately after the oxidation, each library was converted with bisulfite reagent solution, following the bisulfite conversion thermal

cycler program in the instruction. Bisulfite converted DNA was then desulfonated at RT for 5 min, following 2× washing with 70% ethanol. For each library, the optimal amplification was determined using qPCR, in which 1/5th of the libraries was added to amplification master mix containing SYBR Green and run on an Applied Biosystems StepOnePlus Real-Time PCR system for 30 cycles. Relative log-fluorescence vs amplification cycle was plotted out to manually determine the appropriate amplification cycles, selected within the middle to late exponential phase of amplification. Based on the optimization, libraries were amplified between 8 and 13 cycles. Following amplification, libraries were purified with a final 1× clean-up using Agencourt AMPure XP beads (Beckman Coulter, A63881). The quality of the libraries was assessed using an Agilent Bioanalyzer 2100 with the High Sensitivity DNA analysis kit (Agilent, 5067-4626) and concentration was determined using Qubit™ dsDNA HS Assay kit (Invitrogen, Q32854). The libraries were sequenced on NovaSeq S4 (GenomeScan, NL) to obtain minimally 20 million PE150 reads per sample.

## RRBS analysis

Processing of the raw reads was done according to the instructions of RRBS-library preparation kit (Tecan/Nugen). Reads were first trimmed for quality and presence of the adaptor using TrimGalore (v0.6.10) with the following command: "trim_galore --paired -a AGATCGGAAGAGC -a2 AAATCAAAAAAAC", followed by removal of the diversity-adapters using a custom Python script provided by Tecan/Nugen ("-b" flag is specified due to double digestion with both MspI and TaqI). Subsequently, the trimmed reads were aligned to GENCODE mm10GRCm38.p6 using Bismark (v0.24.1) (Krueger and Andrews, 2011) with a maximum insert size of 500 bp. The Tecan/Nugen kit implements a UMI as the last 6 bp of I1 read, thus the aligned reads were deduplicated using UMI-tools (Smith et al, 2017), followed by methylation extraction with the bismark methylation-extractor (v0.24.1). On average, we reached a

coverage of 8.7X and detected $2.6 \times 10^6$ CpGs with a minimal coverage of 5X per library, covering roughly 13% of mm10.

Differentially methylated regions (DMRs) were called based on two different strategies (Mulholland et al, 2020) for every pair-wise comparison using MethylKit (v1.26.0) in R (v4.3.1) (Akalin et al, 2012). First, differentially methylated CpGs were called (q-value < 0.05, Δ5mC < 10%) with a minimal coverage of 5X and were merged together (distance ≤ 250 bp, width ≥ 100 bp). Secondly, differential methylation scores (q-value < 0.05, Δ5mC < 10%) were calculated for covered regions, based on the UCSC annotation of promoters, CpG islands, CpGi shores, CpGi shelves and 1 kb tiles with a 500 bp sliding window, containing at least 3 measured CpGs. From both strategies, regions that were overlapping were merged, filtered for minimally ≥2 differentially methylated CpGs and separated into hyper and hypo methylated DMRs. Per pairwise comparison, we only statistically tested CpGs that have an overall standard deviation of 2% methylation rates within all samples (Akalin et al, 2012). DMRs were annotated using AnnotatR (v1.26.0) and ChIPseeker (v1.36.0). Genes were associated with DMRs using the rGREAT (v2.2.0) package and GO term enrichment was performed using Cluster Profiler (4.8.1). DMR subsets were made using bedtools (v2.31.0). Profile plots were made using DeepTools (v3.5.5).

## Whole genome bisulfite sequencing (WGBS) library preparation

400 ng of genomic DNA was sonicated on a Covaris LE220 with settings targeting an average size of 400 bp. Prior to bisulfite conversion, 0.5% unmethylated Lambda gDNA was spiked in and DNA quantification performed using Picogreen. 400 ng of sheared DNA was bisulfite converted with the Zymo EZ DNA Methylation Gold kit (cat# D5005, VWR) following the product manual. Estimating a 75% recovery rate, 50 ng of deaminated DNA was used for library prep with the IDT xGen™ Methyl-Seq DNA library prep kit (cat#10009824). Briefly, gDNA was heat-denatured and the single-stranded DNA was used as template for proprietary adaptase stub tailing and adapter ligation prior to full-length adapter addition during the indexing PCR amplification of the library. Following product purification, the resulting libraries were quantitated by Picogreen and fragment size assessed with the Agilent 2100 Bioanalyzer. All samples were pooled equimolarly and re-quantitated by qPCR using the Applied Biosystems ViiA7 Quantitative PCR instrument and a KAPA Library Quant Kit (p/n KK4824). Using the concentration from the ViiA7™ qPCR machine above, 150 pM of equimolarly pooled library was loaded onto one lane of the NovaSeq S4 flowcell (Illumina p/n 20028312) following the XP Workflow protocol (Illumina kit p/n 20043131) and amplified by exclusion amplification onto a nanowell-designed, patterned flowcell using the Illumina NovaSeq 6000 sequencing instrument. PhiX Control v3 adapter-ligated library (Illumina p/n FC-110-3001) was spiked-in at 2% by weight to ensure balanced diversity and to monitor clustering and sequencing performance. A paired-end 150 bp cycle run was used to sequence an average of 400 million read pairs (~40X coverage) per sample.

## WGBS analysis

TrimGalore (v0.6.10) was used to trim reads based on quality (PHRED < 30) and to remove adapter sequences. Further, from both read 1 and read 2, 15 bp were trimmed from the 5′ end and 5 bp from the 3′ end. Only reads with a minimum length of 20 bp were kept. Using Bismark (v0.23.1), the trimmed reads were aligned to GENCODE mm10 GRCm38.p6 with a maximal insert size of 500 bp, followed by deduplication and methylation extraction. Using a custom script the CpG counts were merged to one strand. The DSS package was used, using a standard workflow, to call differentially methylated loci, which were used to find differentially methylated regions (DMRs). We kept DMRs with a minimal difference of 10% in methylation, that contained minimal 3 CpGs and had a minimal length of 100 bp. DMRs were annotated using AnnotatR (v1.16.0) and ChIPseeker (v1.26.2) packages in R (v4.0.5) using UCSC gene feature and CpG island (CpGi) locations. Genes were associated with DMRs using the rGREAT (v1.22.0) package. GO term enrichment was performed using Cluster Profiler (v3.18.1).

## RNA-seq library preparation

From embryonic brain tissues, RNA was extracted using RNeasy plus mini kit (Qiagen, 74136) according to the manufacturer's instructions. The integrity of RNA was confirmed using Agilent Bioanalyzer 2100 with RNA 600 Nano kit (Agilent, 5067-1511). Libraries were prepared from 1 µg of total RNA using the KAPA Stranded mRNA Hyperprep Kit (96rxn) (Roche, KK8581) according to manufacturer's specifications. 7 µM KAPA-single index adapters (Roche, KK8700d) were added to A-tailed cDNA, and libraries were amplified for 10 cycles. Finally, 1x library clean-up was performed using Agencourt AMPure XP beads (Beckman Coulter, A63881). Library fragment size was assessed using Agilent Bioanalyzer 2100 with the High Sensitivity DNA analysis kit (Agilent, 5067-4626) and concentration was determined using Qubit™ dsDNA HS Assay kit (Invitrogen, Q32854). Each library was diluted to 4 nM and pooled for sequencing on an Illumina Hiseq4000 at KU Leuven Genomics Core (Belgium), aiming at 15–20 million SE50 reads per sample (19 million reads on average).

## RNA-seq analysis

Adapters, polyA/T tails, and bad quality reads (Phred score > 20) were trimmed using Trim Galore! (v0.6.4_dev) with default parameters. Reads were aligned to the transcriptome and quantified using Salmon (v0.14.1) (Patro et al, 2017) with default parameters using GENCODE release 23 of the mouse reference transcriptome sequences and gene annotation. Subsequently, the counts were imported into R (v4.0.2) using tximport (v1.18.0). Differentially expressed genes were defined using DEseq2 (v1.30.0) (Love et al, 2014) (FDR adjusted p.val < 0.05) and log fold changes corrected using "ashr" method (Stephens, 2017). GO term enrichment was performed using Cluster Profiler (3.18.1). TPM values were calculated using tximport.

## Targeted amplicon bisulfite sequencing library

We assayed the *Slc46a1* promoter methylation status by designing 8 amplicons of 150–290 bp each, spanning 1320 bp of the promoter region, for analysis by targeted bisulfite sequencing as described previously (van der Veer et al, 2023). In brief, 1.5 µg of gDNA was bisulfite-treated using the EpiTect Fast DNA Bisulfite kit (Qiagen, 59824) and eluted in 15 µl of elution buffer. A 20 µl PCR reaction was composed of 0.5 µl of bisulfite-converted gDNA input, 300 nM each of forward and reverse primers (containing P7 and P5 tails, respectively, Table 3), Platinum™ Taq DNA polymerase High Fidelity (Invitrogen, 11304-011) and PCR kit buffers. PCR products were gel-extracted using the PureLink Quick Gel Extraction kit (Invitrogen, K210012). Amplicon concentrations were quantified with the Qubit™ dsDNA HS Assay kit (Invitrogen, Q32854). Amplicons were diluted to 15 nM per sample, pooled to a maximum concentration of 5 ng/µl and assessed for DNA quality using a fragment analyzer (Agilent) and the Qubit™ dsDNA HS Assay kit (Invitrogen, Q32854). Amplicon pools were converted into sequencing libraries by performing a secondary PCR to incorporate indexes and sequencing adapters. The PCR reaction included 9 µl of DNA, 0.5 µl of custom p7 primer (125 nM), 0.5 µl of custom p5 primer (125 nM), and 1 × 10 µl Phusion® High Fidelity PCR master Mix with HF buffer (Biolabs new England M0531S). The thermocycler program was 94 °C for 30 s; 15 cycles of 94 °C for 10 s, 51 °C for 30 s, 72 °C for 30 s; and a final extension at 72 °C for 1 min. Custom primers were provided with a unique dual index for sample labeling. After purification using AMPure XP beads following the manufacturer's protocol, the library's final quality was analyzed with a fragment analyzer and then pooled equimolarly. The concentration of the final pool was measured using qPCR (Kapa SYBR fast, Roche, KK4600), diluted to 4 nM and loaded onto a NovaSeq for PE150 sequencing, aiming for a minimum of 200,000 reads per amplicon and an average of 350,000 reads. For data analysis, reads were processed using Trim Galore! (v0.6.7) for quality, adapter removal, and exclusion of reads shorter than 20 bp. Subsequently, Bismark (v0.23.1) aligned the trimmed reads to GENCODE mm10 GRCm38.p6, with a maximal insert size of 500 bp, followed by methylation extraction. Only CpGs with a minimum coverage of 1000x were retained for analysis and were visualized over the genomic locus using a custom R script (v4.0.3).

## Data availability

RNA-seq, RRBS and WGBS datasets produced in this study were deposited to Gene Expression Omnibus (GEO): GSE275393, GSE275401 and GSE276883, respectively. All other data are available in the main text or the supplementary materials.

The source data of this paper are collected in the following database record: biostudies:S-SCDT-10_1038-S44319-024-00316-1.

## Peer review information

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

## Acknowledgements

RNA-seq libraries were sequenced by Genomics Core at KU Leuven. We thank Wouter Bossuyt, David Carbonez, and Annelien Verfaillie at the Core for their excellent technical support in NGS experimental design and bioinformatic analysis. We thank Sarah Cornet and Kelly Nguyen, our internship students, for their help in collecting the amplicons for *Slc46a1* bisulfite amplicon sequencing and Rita San-Bento at Tecan (Lyon, France) for technical guidance on RRBS library preparation. WGBS libraries were prepared and sequenced by Emily Ricco, Ping Kang, and Daniel C. Kraushaar at the Genomic and RNA Profiling Core at Baylor College of Medicine. We thank Steffen Fieuws at the Leuven Biostatistics and Statistical Bioinformatics Centre for advice on statistical analysis. This work was supported by the Belgium Research Foundation – Flanders (FWO) Research Projects G092518N (KPK), G0C6820N (KPK), and

KU Leuven Internal Funds C14/21/117 (KPK) and a U.S. National Institute of Environmental Health Sciences Gulf Coast Center for Precision Environmental Health pilot award P30ES030285 (KPK). BKV is a recipient of FWO PhD fellowship 11E7920N and KU Leuven postdoctoral mandate PDMT2/23/083. LC is supported by a PhD scholarship No. 202004910440 from China Scholarship Council. SCT is a recipient of the MSCA SoE FWO postdoctoral fellowship 673343/12ZZE23N. The computational resources and services used in this work were provided by the VSC (Flemish Supercomputer Center), funded by FWO and the Flemish Government. Dr. Finnell previously held a leadership position in TeratOmic Consulting, LLC. He also received travel funds to attend editorial board meetings of the journal Reproductive and Developmental Medicine. The content is solely the responsibility of the authors and does not necessarily represent the official views of the National Institutes of Health.

## Author contributions

**Lehua Chen**: Conceptualization; Data curation; Formal analysis; Validation; Investigation; Visualization; Methodology; Writing—original draft; Project administration; Writing—review and editing. **Bernard K van der Veer**: Conceptualization; Data curation; Formal analysis; Validation; Investigation; Visualization; Methodology; Writing—original draft; Project administration; Writing—review and editing. **Qiuying Chen**: Data curation; Formal analysis; Validation; Methodology. **Spyridon Champeris Tsaniras**: Formal analysis; Visualization. **Wannes Brangers**: Investigation. **Harm H M Kwak**: Investigation. **Rita Khoueiry**: Investigation; Methodology. **Yunping Lei**: Methodology. **Robert Cabrera**: Methodology. **Steven S Gross**: Resources. **Richard H Finnell**: Resources; Funding acquisition. **Kian Peng Koh**: Conceptualization; Resources; Supervision; Funding acquisition; Writing—original draft; Writing—review and editing.

Source data underlying figure panels in this paper may have individual authorship assigned. Where available, figure panel/source data authorship is listed in the following database record: biostudies:S-SCDT-10_1038-S44319-024-00316-1.

## Disclosure and competing interests statement

The authors declare no competing interests.

