## [Peer Review File · EMBO Reports]

The DNA demethylase TET1 modifies the impact of maternal folic acid status on embryonic brain development.

Lehua Chen, Bernard van der Veer, Qiuying Chen, Spyridon Tsaniras, Wannas Brangers, Harm Kwak, Rita Khoueiry, Yunping Lei, Robert Cabrera, Steven Gross, Richard Finnell, and Kian Peng Koh

Corresponding author(s): Kian Peng Koh (kian.koh@kuleuven.be)

Review Timeline:

Submission Date:	15th May 24
Editorial Decision:	21st Jun 24
Revision Received:	3rd Sep 24
Editorial Decision:	23rd Oct 24
Revision Received:	28th Oct 24
Accepted:	30th Oct 24

Editor: Deniz Senyilmaz Tiebe

Transaction Report:

Dear Dr. Koh,

Thank you for transferring your research manuscript to our journal, which was now seen by three referees, whose reports are copied below.

Referees express interest in the proposed role of TET1 in regulation of the impact of maternal folic acid availability on emergence of neural tube defects in the embryo. However, they also raise significant concerns that need to be addressed to consider publication here.

I agree with referee #2 that establishing the epistatic relationship between Tet1 and Slc46a1 would significantly strengthen the manuscript (referee #3, point 3). However, I can also understand that this could be technically challenging. Please let me know if you would like to discuss this point further.

Given these positive recommendations, we would like to invite you to submit a revised manuscript. Please revise your manuscript with the understanding that the referee concerns (as in their reports) must be fully addressed and their suggestions taken on board. Please address all referee concerns in a complete point-by-point response. Acceptance of the manuscript will depend on a positive outcome of a second round of review. It is EMBO reports policy to allow a single round of major experimental revision only and acceptance or rejection of the manuscript will therefore depend on the completeness of your responses included in the next, final version of the manuscript.

We realize that it is difficult to revise to a specific deadline. In the interest of protecting the conceptual advance provided by the work, we recommend a revision within 3 months. Please discuss the revision progress ahead of this time with me if you require more time to complete the revisions, or if you have questions or comments regarding the revision (also by video chat).

1. A data availability section providing access to data deposited in public databases is missing (where applicable).
2. Your manuscript contains statistics and error bars based on $n=2$. Please use scatter plots in these cases.

You can submit the revision either as a Scientific Report or as a Research Article. For Scientific Reports, the revised manuscript can contain up to 5 main figures and 5 Expanded View figures, and it should not exceed 27000 characters. If the revision leads to a manuscript with more than 5 main figures it will be published as a Research Article. In this case the Results and Discussion section should be separate. If a Scientific Report is submitted, these sections have to be combined. This will help to shorten the manuscript text by eliminating some redundancy that is inevitable when discussing the same experiments twice. In either case, all materials and methods should be included in the main manuscript file.

3) We replaced Supplementary Information with Expanded View (EV) Figures and Tables that are collapsible/expandable online. A maximum of 5 EV Figures can be typeset. EV Figures should be cited as 'Figure EV1, Figure EV2' etc... in the text and their respective legends should be included in the main text after the legends of regular figures.

4) a .docx formatted letter INCLUDING the reviewers' reports and your detailed point-by-point responses to their comments. As part of the EMBO publication's Transparent Editorial Process, EMBO reports publishes online a Review Process File (RPF) to

accompany accepted manuscripts. This File will be published in conjunction with your paper and will include the referee reports, your point-by-point response and all pertinent correspondence relating to the manuscript.

<https://www.embopress.org/page/journal/14693178/authorguide#transparentprocess>

5) a complete author checklist, which you can download from our author guidelines

<https://www.embopress.org/page/journal/14693178/authorguide>. Please insert information in the checklist that is also reflected in the manuscript. The completed author checklist will also be part of the RPF.

6) Please note that all corresponding authors are required to supply an ORCID ID for their name upon submission of a revised manuscript (<<https://orcid.org/>>). Please find instructions on how to link your ORCID ID to your account in our manuscript tracking system in our Author guidelines

<<https://www.embopress.org/page/journal/14693178/authorguide#authorshipguidelines>>

Additional information on source data and instruction on how to label the files are available:

<https://www.embopress.org/page/journal/14693178/authorguide#sourcedata>

9) Our journal encourages inclusion of *data citations in the reference list* to directly cite datasets that were re-used and obtained from public databases. Data citations in the article text are distinct from normal bibliographical citations and should directly link to the database records from which the data can be accessed. In the main text, data citations are formatted as follows: "Data ref: Smith et al, 2001" or "Data ref: NCBI Sequence Read Archive PRJNA342805, 2017". In the Reference list, data citations must be labeled with "[DATASET]". A data reference must provide the database name, accession number/identifiers and a resolvable link to the landing page from which the data can be accessed at the end of the reference. Further instructions are available at <http://www.embopress.org/page/journal/14693178/authorguide#referencesformat>

10) Regarding data quantification (see Figure Legends:

<https://www.embopress.org/page/journal/14693178/authorguide#figureformat>)

11) The journal requires a statement specifying whether or not authors have competing interests (defined as all potential or actual interests that could be perceived to influence the presentation or interpretation of an article). In case of competing

interests, this must be specified in your disclosure statement. Further information: <https://www.embopress.org/competing-interests>

12) Please also note our reference format:

13) All Materials and Methods need to be described in the main text. We would encourage you to use 'Structured Methods', our new Methods format. According to this format, the Methods section should include a Reagents and Tools Table (listing key reagents, experimental models, software and relevant equipment and including their sources and relevant identifiers) followed by a Methods and Protocols section in which we encourage the authors to describe their methods using a step-by-step protocol format with bullet points, to facilitate the adoption of the methodologies across labs. More information on how to adhere to this format as well as downloadable templates (.doc or .xls) for the Reagents and Tools Table can be found in our author guidelines: <https://www.embopress.org/page/journal/14693178/authorguide#manuscriptpreparation>.

An example of a Method paper with Structured Methods can be found here: <https://www.embopress.org/doi/full/10.1038/s44320-024-00037-6#sec-4>

I look forward to seeing a revised version of your manuscript when it is ready. Please let me know if you have questions or comments regarding the revision.

Kind regards,

Deniz Senyilmaz Tiebe

Deniz Senyilmaz Tiebe, PhD
Scientific Editor
EMBO Reports

Referee #1:

Chen and van der Veer et al. report on the combined effects of Tet1 (Tet Methylcytosine Dioxygenase 1) mutation, genetic background, and dietary folic acid (FA) intake on neural tube defects (NTD), phospholipid metabolite status, and DNA methylation levels in mice. TET1 is a dioxygenase that regulates active DNA demethylation and by extension gene expression. This team has already a substantive and excellent track record in working with TET functions in embryonic development and the current manuscript presents a continuation of their earlier work in the field.

The paper presents a wealth of new data by focusing on the analysis of previously generated B6-Tet1^{tm1Koh} mice. The authors confirm that Tet1 loss leads to NTD in a background dependent manner with CD1 outbred mice showing greater rates of NTD than C57BL/6 congenic mice. Outcross to 129 background leads to a parabolic effect on NTD rates with 5-9 backcross generations resulting in higher rates and {greater than or equal to} 13 backcross generations in lower rates of NTD. These peculiar findings remain unexplained. CD1 NTD rates remain also unaffected by FA or methionine oversupplementation, but metabolic profiles, probably unsurprisingly, change substantially depending on genotype and FA status.

Subsequently, the authors generate three dietary groups supplied with .1, 3, and 30 ppm FA, while folate production by the gut microbiome is suppressed by the addition of 1% succinyl sulfathiazole (SST). They proceed to analyze E11.5 offspring of these dietary groups on WT, het, or homozygous Tet1 backgrounds with respect to outcomes mentioned above. This carefully constructed study design yields a great amount of data that are comprehensively collected and analyzed to the highest standards, covering phenotypic, metabolomic, methylomic, and transcriptomic work. There is not much to fault here, except for the choice of developmental stage to analyze. If the goal of this study was to establish a molecular connection between NTD and TET1/folate status, the choice should have been to collect embryos for molecular analyses around the time of neural tube closure, ~E9.0. The DNA methylome and gene expression are subject to highly dynamic temporal regulation and appropriate timing seems appropriate to me. Please clarify and justify, why E11.5 embryo were chosen for this work.

A separate problem of this study is the lack of cohesion. Data are being presented without leading to any mechanistic insight or being meaningfully connected to each other. This lack of cohesion is exacerbated by often convoluted and near impenetrable writing that makes it very difficult to follow rationale and outcomes and tie them to any overarching theme. While these are not necessarily grounds to reject this manuscript, the authors would help themselves and their prospective readership in simplifying some of their sentences and potentially also omit data that may be extraneous to the core aspect of this work. In addition, some

statements seem to contradict their results. The problems start already with the abstract, in line 25: "Both excess and depleted maternal FA diets interact with loss of Tet1 to alter offspring DNA methylation primarily at neurogenesis loci." However, Fig. 4 suggests that DMRs associated with Tet1 loss primarily affect mesenchymal systems and FA x TET1 DMRs primarily axonogenesis, and neurogenesis appears only as the 5th GO term. Moreover, the much more interesting fact, due to its functional implications, that differential gene expression upon combined Tet1 loss and FA excess (Fig.5) seem to center around synaptic function is omitted here. Also, line 27: "Excess FA in Tet1 deficient embryos results in reduced FA intake..." How can excess FA intake result in reduced FA intake? Please clarify, as it was impossible to retrieve from the provided text. The next sentence reads: "Conversely, FA deficiency reveals NTD susceptibility due to haploinsufficiency of Tet1 in inbred strains." But no converse results are being presented here, as separate outcomes are being discussed. The final sentence of the abstract reads: line 31: "Overall, our study unravels interactions between modified maternal FA status and Tet1 gene dosage that impact neurotransmitter functions and cellular methylation, implicating epigenetic dysregulation as a mechanistic basis for NTDs resistant to FA supplementation." But no mechanistic connection between methylation status and NTD prevalence has been provided throughout the manuscript. Line 260: "In our sample sets of 54 embryos (Supplementary Table S1), we intentionally included a few WT and HET embryos with mild deformities (but stage-matched based on Theiler stages) and selected against KO embryos with severe malformation." Why? Another example would be: line 303 "Our final sample set included a total of 36 embryos from 5 biological replicates, male and female, representative of the phenotypes observed per group." In my mind, a biological replicate is an individual embryo. How can 36 embryos amount to 5 biological replicates? Please clarify.

The final element of the paper assessing Slc46a1 methylation and expression seems out of place. First, the authors confirm that "TPM values of Slc46a1, Slc19a1, Fcrl1 and Fcrl2 in our RNA-seq analysis of 129S6.Cg E11.5 brains did not show any significant changes in expression of these genes affected by Tet1 genotype and FA diet." Undeterred, they move the question of epigenetic regulation to an in vitro system by performing in vitro differentiation of WT and Tet1 KO ESCs into anterior neural progenitor cells where they observe somewhat reduced expression of Slc46a1. How this approach is of any relevance to in vivo development of the mouse embryo and even less so to human NTD remains unclear and this question could have been more directly investigated in mouse embryos by any number of methods.

Below are a few minor comments:

I was not able to find supplementary table 1.

The manuscript contains occasional grammatical errors and should be proofread again.

Referee #2:

The authors conducted a comprehensive analysis examining the role of genetic heterogeneity, Tet1 knockout, and folic acid (FA) intake in neural tube closure defects. Initially, they observed that Tet1 knockout-induced neural tube closure defects were prevalent in genetically heterogeneous mouse strains. Additionally, they found that these defects were not ameliorated by FA ingestion. Omics analyses of metabolism, DNA methylation, and gene expression revealed various alterations under certain conditions. Notably, altered expression and DNA methylation were detected in Slc46a1, a gene encoding an FA transporter.

While the authors have performed extensive omics analyses across multiple conditions, it remains unclear how the findings from these analyses contribute to the understanding of neural tube closure defects. Several critical issues must be addressed before this manuscript can be considered for publication.

Major Points:

1. The relationship between Tet1 knockout and FA intake requires further clarification. The contradictory effects of FA observed in Figures 2B and 2D suggest that the relationship between Tet1 and FA may vary across different strains. Even if the findings in Figures 3, 4, 5, and 6 are valid for the 129S6 strain, it is essential to determine if these results are also applicable to the CD1 strain.
2. The results from the metabolome, DNA methylome, and RNA-seq analyses do not clearly elucidate the mechanisms underlying neural tube closure defects caused by Tet1 knockout and FA intake. For instance, the authors demonstrated alterations in metabolites involved in lipid metabolism, changes in DNA methylation of genes related to nervous system development, and reduced expression of genes involved in neurotransmission. However, since nervous system development and neurotransmission mainly occur post-neural tube closure, the relevance of these findings to neural tube closure remains unclear. The omics results should be discussed in direct relation to the neural tube closure phenotype.
3. The involvement of Slc46a1 in neural tube closure needs further investigation. It should be determined whether Slc46a1 knockout causes neural tube closure defects and if overexpression of Slc46a1 can rescue the Tet1 knockout phenotype.
4. While pairwise comparisons were performed in Figures 4 and 5, a multiple comparison test among the nine groups is necessary to examine overlaps in Figures 4D and 5E.
5. The color scheme in Figures 1H and 3F is not distinguishable for colorblind individuals, such as this reviewer. They should use the same coloring scheme as in Figure 5C to improve accessibility.

Minor Points:

6. Figure 1A should include the Control genotype for comparison.
7. All replicates should be shown in Figure 3E, rather than presenting only the average.
8. The terms "Tet1-specific" and "FA x TET1" in Figure 4D are misleading. It may be clearer to use "Tet1-dependent and FA-independent" and "both Tet1- and FA-dependent," respectively.
9. On page 19, it is suggested that excess FA in the absence of TET1 may mimic FA deficiency. According to Figures 3B-D, the levels of folic acid, SAM/SAH, and Serine/Glycine differ between the 30 ppm and 0.1 ppm FA groups. Additionally, the CpG level in the promoter of Slc46a1 (Figure 6E) exhibits different patterns between the 30 ppm and 0.1 ppm FA groups. The mechanisms by which differing FA availability leads to opposite downstream effects should be elucidated.
10. According to Figure 4B, the normal diet (3 ppm FA) is the most effective condition for hyperDMRs, whereas Figure 2D suggests that FA deficiency has the most significant impact on the phenotype. Furthermore, in Figure 5A, the normal diet appears to have a limited effect on DEGs. The significance of hyperDMRs needs to be clearly explained in this context.

Referee #3:

1. Does this manuscript report a single key finding? YES. Loss of TET1 function leads to blunted molecular response to maternal FA supplementation, thus Tet1^{-/-} embryos resemble NTD cases that are non-responsive to FA-supplementation.
2. Is the reported work of significance (YES), or does it describe a confirmatory finding or one that has already been documented using other methods or in other organisms etc (NO)? YES
3. Is it of general interest to the molecular biology community? YES, this study employs a lot of controls and extensive pair-wise comparison in order to distinguish phenotypes (molecular and developmental phenotypes) that are caused by 1) environmental exposure, 2) genotypes, and 3) interactions between exposure and genotypes.
4. Is the single major finding robustly documented using independent lines of experimental evidence (YES), or is it really just a preliminary report requiring significant further data to become convincing, and thus more suited to a longer-format article (NO)? YES

In this manuscript, Chen et al. sought to understand how maternal folic acid (FA) levels influence embryonic brain development, specifically neural tube closing, in the context of TET1 loss of function. This study is partly motivated by high numbers of human neural tube defects (NTDs) that are resistant to FA supplement. As folate feeds into the one-carbon metabolism pathway that produces the methyl donor S-adenosylmethionine, the authors posit that interactions between genetic and the maternal FA environment are facilitated through the 5mC dioxygenase TET1. TET1 loss of function mouse models show varying degrees of NTD penetrance that is dependent on the homogeneity of the genetic background of the mouse line. The authors spent considerable effort to characterize the baseline NTD rates in the highly penetrant CD1-Tet1^{-/-} line and the less-sensitive, but more congenic (thus, more appropriate for downstream molecular analyses) 129S6-Tet1^{-/-} line. Overall, the authors conclude that the loss of TET1 renders embryos less responsive to molecular changes elicited by FA supplementation. This may be explained by the blunted delivery of excess FA to 129S6-Tet1^{-/-} embryos due to decreased expression of Slc46a1. Interestingly, FA-deficiency does not seem to exacerbate the NTD or molecular phenotypes of 129S6-Tet1^{-/-} embryos. I have a couple major comments that need to be addressed and several minor comments that will help to streamline the various models used in this study which I have included below:

Major comments:

1. Beginning in Figure 2, there are significant discussions on the effect of Tet1 haploinsufficiency on the responsiveness to maternal FA levels, including a stark subset of HET-specific DEGs in response to FA-deficiency. In this manuscript or the preceding Khoueiry et al., 2017 where the Tet1^{tm1Koh} was initially generated, I cannot find mRNA or protein levels of WT vs HET vs KO to support that 1) no full-length protein is generated in the KO or 2) TET1 is actually present at reduced level in 129S6-Tet1^{+/-} embryos or embryonic brains. In fact, as the authors pointed out in Supplementary Figure 5B, this gene-trap allele still allows for expression of the short somatic Tet1 isoform from a downstream TSS in the KO embryonic brains. A careful validation of the gene product of the Tet1^{tm1Koh} allele should be included, including a Western blot using both C- and N-terminus TET1 antibody, preferably on embryonic brains of relevant developmental stage. In relation to this point, have the authors checked expression levels of Tet1 (from both the canonical TSS and the downstream alternative TSS) in Tet1^{-/-} as the line was outcrossed to CD1 and as the line was backcrossed to the 129S6 background (in generations where NTD was high in penetrance N5-9 and when the 129S6 was considered congenic). Is there a correlation between expression of the short isoform of Tet1 (if the full length Tet1 is indeed absent) and the percentage of KO-embryos with NTD?
2. The RRBS was conducted using oxidative bisulfite (ox-BS) pipeline. This should allow for delineation of 5mC vs 5hmC within the library through subtractive analysis methods if the BS-libraries were sequenced in parallel. I will preface this comment by mentioning that I recognize the low-level of 5hmC in the embryonic brain and that most 5hmC accumulation occurs postnatally in post-meiotic neurons. In this study, the dietary (excess or deficient FA) effects on the methylome, and to some degree gene expression, is only observed in the WT and HET, and not in the KO (Figure 4C, 5D right panel). This suggests that TET1's activity (5hmC generation) facilitates response to differing levels of maternal FA. Additionally, since there is very little relationship

between DNA methylome changes to the observed changes in gene expression, 5hmC profiling should be conducted to obtain a comprehensive view on the interactions between 1-C metabolism, Tet1 genetic status, and NTD phenotype. While the literature is scant, there are emerging findings that NTDs are associated with global decreased in global 5hmC levels (PMIDs: 36199572; 31179819).

Minor points:

1. As a reader, I consider CD1-Tet^{-/-} vs 129S6-Tet1^{-/-} to be completely different mouse lines when I interpret the result. While in many figures the genetic backgrounds are included in the labeling, they are missing in several of them and only included in the figure legends. Please consider always including the genetic background in labeling the figures (e.g. Figure 1C, 1H, Figure 3, 4, 5, and 6).
2. Have the authors measured FA metabolites/FA levels in serum of the dam?
3. Line 83 should read: "...DNA demethylation via the iterative oxidation..."
4. Line 128, should E10.5 be E9.5 instead as that is the time point analyzed in Figure 1C.
5. Please refer to the strain of KO in line 244.
6. Figure 3B should be ordered the same way as Figure 3C and 3D.
7. Figure 3F: I think the ordering of this heatmap should be changed in order to communicate the result better. I think the authors are trying to relay that the FA-dose effect is seen in WT and HET but not in KO. In order to immediately see this dose-dependent effect on metabolite the top layer of ordering should be genotype, then intercalate the dose as the second layer.
8. Related to Figure 4A, can the authors provide violin plots of global methylation levels to complement these metagene plots so that readers can quickly assess changes in global methylation levels between genotypes and diets.
9. Related to Figure 4B, are there overlap between the DMRs found in KO vs WT (3ppm FA) and those found in HET vs WT (3ppm FA)? Where in the genome are these overlapping DMRs located in?
10. Related to Figure 4C, there are many DMRs that are found in WT in response to excess or deficient maternal FAs. What are the GO terms for these exclusively FA-driven DMRs and are there specific genomic compartments that these FA-driven DMRs located in?
11. I do not agree with the rationale for DEGs overlapping in Figure 5E. I do not think this is informative to identify "genes that are responding to dietary FA fluctuations". Instead of Figure 5E and 5F, can the authors instead provide deepTools analyses of where the DMRs are located with respect to DEGs (distance from DEGs, or metaplots of where in DEGs). Or alternatively, where TET1 is found (from the TET1 EpiLC ChIPseq for example) with respect to DEGs.
12. For ESC to antNPC differentiation, the authors mention in line 450 of using "defined culture media containing high FA supplements (>5mg/L)". Is this concentration of FA supplement a requirement for antNPC differentiation or is it a culture condition that is supposed to mimic high maternal FA environment?

We thank the reviewers for their comments and suggestions which we have used to dramatically improve the manuscript. Please see our point-by-point response below. Black=Reviewer Comment. Blue=Response.

Referee #1:

Chen and van der Veer et al. report on the combined effects of Tet1 (Tet Methylcytosine Dioxygenase 1) mutation, genetic background, and dietary folic acid (FA) intake on neural tube defects (NTD), phospholipid metabolite status, and DNA methylation levels in mice. TET1 is a dioxygenase that regulates active DNA demethylation and by extension gene expression. This team has already a substantive and excellent track record in working with TET functions in embryonic development and the current manuscript presents a continuation of their earlier work in the field.

The paper presents a wealth of new data by focusing on the analysis of previously generated B6-Tet1tm1Koh mice. The authors confirm that Tet1 loss leads to NTD in a background dependent manner with CD1 outbred mice showing greater rates of NTD than C57BL/6 congenic mice. Outcross to 129 background leads to a parabolic effect on NTD rates with 5-9 backcross generations resulting in higher rates and {greater than or equal to} 13 backcross generations in lower rates of NTD. These peculiar findings remain unexplained. CD1 NTD rates remain also unaffected by FA or methionine oversupplementation, but metabolic profiles, probably unsurprisingly, change substantially depending on genotype and FA status.

Response: The impact of genetic modifiers on NTD penetrance in *Tet1* KO embryos and the identification of a strong quantitative trait locus based on the intercrossing between B6 and 129S6 strains are described in the separate manuscript (see pre-print <https://doi.org/10.1101/2024.02.21.581196>). The higher penetrance of NTDs in *Tet1* KO on non-inbred than on highly inbred genetic backgrounds suggest that complex interactions between strain-specific variants contribute to the phenotype.

Subsequently, the authors generate three dietary groups supplied with .1, 3, and 30 ppm FA, while folate production by the gut microbiome is suppressed by the addition of 1% succinyl sulfathiazole (SST). They proceed to analyze E11.5 offspring of these dietary groups on WT, het, or homozygous Tet1 backgrounds with respect to outcomes mentioned above. This carefully constructed study design yields a great amount of data that are comprehensively collected and analyzed to the highest standards, covering phenotypic, metabolomic, methylomic, and transcriptomic work. There is not much to fault here, except for the choice of developmental stage to analyze. If the goal of this study was to establish a molecular connection between NTD and TET1/folate status, the choice should have been to collect embryos for molecular analyses around the time of neural tube closure, ~E9.0. The DNA methylome and gene expression are subject to highly dynamic temporal regulation and appropriate timing seems appropriate to me. Please clarify and justify, why E11.5 embryo were chosen for this work.

Response: In retrospect, we agree with Reviewer#1 that E8.5-9.5 would be a more appropriate time-point to relate the epigenomic and metabolomic changes to cranial neural tube closure (NTC) in the mouse. This manuscript is a result of a preliminary assessment of FA-responsiveness of NTDs and other cranial phenotypes (requiring analyses post-NTC) in different strains of *Tet1* KO mice. Because the NTDs were not FA-responsive, we changed our focus to embryonic brain development by analyzing all E11.5 embryos collected from the phenotyping studies, without any prior expectation that the results will relate to NTC. The readout we focus on is whether FA status modifies the impact of TET1 on DNA

methylation. The results provide strong preliminary data to support our next steps to investigate this phenomenon in E8.5 headfold tissues.

A separate problem of this study is the lack of cohesion. Data are being presented without leading to any mechanistic insight or being meaningfully connected to each other. This lack of cohesion is exacerbated by often convoluted and near impenetrable writing that makes it very difficult to follow rationale and outcomes and tie them to any overarching theme. While these are not necessarily grounds to reject this manuscript, the authors would help themselves and their prospective readership in simplifying some of their sentences and potentially also omit data that may be extraneous to the core aspect of this work. In addition, some statements seem to contradict their results. The problems start already with the abstract, in line 25: "Both excess and depleted maternal FA diets interact with loss of Tet1 to alter offspring DNA methylation primarily at neurogenesis loci." However, Fig. 4 suggests that DMRs associated with Tet1 loss primarily affect mesenchymal systems and FA x TET1 DMRs primarily axonogenesis, and neurogenesis appears only as the 5th GO term.

Response: We hope the revisions in Figures 4 and 5 in response to specific comments by the other reviewers addresses the apparent lack of cohesion. In this revision, we have consulted with a biostatistician and taken caution to use the term "interact" only in statements supported by two-way ANOVA analysis of diet x genotype interactions. The problematic sentence in the abstract has been revised to: "Maternal diets that are either too rich or deficient in FA resulted in an increased incidence of mild cranial deformities in wild type and Tet1 heterozygous offspring and altered DNA hypermethylation in nullizygous embryos primarily at neurodevelopmental loci." While "axonogenesis" is indeed the top GO term enriched by hyper DMRs in the TET1 x FA diet-specific, "regulation of neurogenesis", "neuron projection guidance" and "axon guidance" terms are also highly significant terms (Figure 4F). To fit the word limit in the abstract, we thought "neurogenesis" would be a generalized word to describe all these terms. To avoid misleading the reader, we have replaced the word with "neurodevelopment." In the main text of Results, we have specified these neuron-related GO terms verbatim for clarity.

Moreover, the much more interesting fact, due to its functional implications, that differential gene expression upon combined Tet1 loss and FA excess (Fig.5) seem to center around synaptic function is omitted here.

Response: We agree with Reviewer 1 and have added text stating that several GO terms related to synaptic function and organization are shown in Figures 5B and 5G.

Also, line 27: "Excess FA in Tet1 deficient embryos results in reduced FA intake..." How can excess FA intake result in reduced FA intake? Please clarify, as it was impossible to retrieve from the provided text.

Response: We have rephrased this phrase to: "Excess FA in *Tet1*^{-/-} embryos caused promoter DNA hypermethylation, reduced expression of multiple membrane solute carriers including a FA transporter, and phospholipid metabolite loss, mimicking a pseudo-FA deficiency."

The next sentence reads: "Conversely, FA deficiency reveals NTD susceptibility due to haploinsufficiency of Tet1 in inbred strains." But no converse results are being presented here, as separate outcomes are being discussed.

Response: We agree and have changed the word "Conversely" to "In addition."

The final sentence of the abstract reads: line 31: "Overall, our study unravels interactions between modified maternal FA status and Tet1 gene dosage that impact neurotransmitter functions and cellular methylation, implicating epigenetic dysregulation as a mechanistic basis for NTDs resistant to FA supplementation." But no mechanistic connection between methylation status and NTD prevalence has been provided throughout the manuscript.

Response: We have revised this concluding sentence to: "Overall, our study reveals the target loci implicated in neurotransmitter functions and suggests that epigenetic dysregulation may underlie NTDs resistance to FA supplementation." Indeed, we are not drawing a mechanistic connection between DNA methylation status and NTD prevalence, because the data show that the metabolomic and methylation changes occur largely independently of the presence or absence of an NTD phenotype. However, we have performed stringent two-way ANOVA to suggest that *Tet1* genotype and FA diet interactions can affect phospholipid metabolism and membrane ion transport expression, fitting a mechanism for resistance to FA supplementation.

Line 260: "In our sample sets of 54 embryos (Supplementary Table S1), we intentionally included a few WT and HET embryos with mild deformities (but stage-matched based on Theiler stages) and selected against KO embryos with severe malformation." Why?

Response: The inclusion of WT and HET embryos with mild deformities is to ensure that the samples are representative of the range of phenotypes observed. The exclusion of KO embryos with severe malformation is because most of these embryos could no longer be TS stage-matched with WT controls. We have revised the sentence accordingly: "In our sample sets of 54 embryos (Expanded View Table S1), we intentionally included a few WT and HET embryos with mild deformities (but stage-matched based on Theiler stages) to ensure that the sample are representative of the range of phenotypes observed. By the same criteria, we excluded KO embryos with severe malformation when they could no longer be TS stage-matched with WT controls, minimizing confounding DNA methylation changes due to gross developmental delays."

Another example would be: line 303 "Our final sample set included a total of 36 embryos from 5 biological replicates, male and female, representative of the phenotypes observed per group." In my mind, a biological replicate is an individual embryo. How can 36 embryos amount to 5 biological replicates? Please clarify.

Response: We thank Reviewer 1 for pointing out this confusing sentence, which also has an error in counting. As explained in the manuscript, we considered embryos from a litter to constitute one biological replicate. We have corrected the sentence to: "Our final sample set included a total of 36 embryos collected from four litters in each of the three diet groups, representative of embryo phenotypes and both sexes."

The final element of the paper assessing *Slc46a1* methylation and expression seems out of place. First, the authors confirm that "TPM values of *Slc46a1*, *Slc19a1*, *Folr1* and *Folr2* in our RNA-seq analysis of 129S6.Cg E11.5 brains did not show any significant changes in expression of these genes affected by *Tet1* genotype and FA diet." Undeterred, they move the question of epigenetic regulation to an in vitro system by performing in vitro differentiation of WT and *Tet1* KO ESCs into anterior neural progenitor cells where they observe somewhat reduced expression of *Slc46a1*. How this approach is of any relevance to in vivo development of the mouse embryo and even less so to human NTD remains unclear

and this question could have been more directly investigated in mouse embryos by any number of methods.

Response: While DNA methylation changes related to NTC may still be detected at post NTC stages because they can persist through development, gene expression changes are much more dynamic. Here, to relate DNA methylation with gene expression changes of folate carrier genes, we speculate that the latter readouts should be at an earlier stage if they are indeed affecting NTC. Unfortunately, the custom FA diets were no longer available in Europe due to new restrictions on the supplementation of antibiotics (SST) in rodent feed and we have no more budget to import the custom diet from alternative suppliers in the US. The only avenue for us is to test this in cell cultures that recapitulate the E8.5 neuroepithelium.

Below are a few minor comments:

I was not able to find supplementary table 1.

Response: We apologize for the missing table, which is now included.

The manuscript contains occasional grammatical errors and should be proofread again.

Response: The revised manuscript has been proof-read by co-authors who are native English speakers.

Referee #2:

The authors conducted a comprehensive analysis examining the role of genetic heterogeneity, Tet1 knockout, and folic acid (FA) intake in neural tube closure defects. Initially, they observed that Tet1 knockout-induced neural tube closure defects were prevalent in genetically heterogeneous mouse strains. Additionally, they found that these defects were not ameliorated by FA ingestion. Omics analyses of metabolism, DNA methylation, and gene expression revealed various alterations under certain conditions. Notably, altered expression and DNA methylation were detected in *Slc46a1*, a gene encoding an FA transporter.

While the authors have performed extensive omics analyses across multiple conditions, it remains unclear how the findings from these analyses contribute to the understanding of neural tube closure defects. Several critical issues must be addressed before this manuscript can be considered for publication.

Major Points:

1. The relationship between *Tet1* knockout and FA intake requires further clarification. The contradictory effects of FA observed in Figures 2B and 2D suggest that the relationship between *Tet1* and FA may vary across different strains. Even if the findings in Figures 3, 4, 5, and 6 are valid for the 129S6 strain, it is essential to determine if these results are also applicable to the CD1 strain.

Response: We do not interpret the effects of FA on the phenotypes observed in Figures 2B and 2D to be contradictory, because the differences in NTD penetrance between *Tet1* KO in different diet groups did not reach statistical significance either in the CD1 or 129S6 backgrounds. A two-sided Chi² test (with alpha=0.05) estimates that scoring 30-50 embryos per group has 80% power to detect a difference of 30-35% in proportion between two groups of equal size. However, these numbers are difficult to reach in our experimental design, because it is not possible to accurately predict how many embryos of each genotype (especially KO) will be obtained in every litter. We are also concerned about the fluctuations of NTD penetrance rates in CD1-*Tet1* KO mice which ranged from 50 to 70% across cohorts (Fig. 2B), likely attributed to the inherent genetic heterogeneity of outbred mice. Therefore, to profile methylome and transcriptome changes, we decided it was more appropriate to perform the genomic analyses in the 129S6 congenic strain, eliminating genetic modifiers as a potential confounder. Nonetheless, the metabolomic analyses involving 2-way ANOVA of genotype and diet factors were performed in CD1, B6 and 129S6.Cg backgrounds. The impact on phospholipid metabolites by *Tet1* genotype- FA diet interactions appear to be consistent across backgrounds.

2. The results from the metabolome, DNA methylome, and RNA-seq analyses do not clearly elucidate the mechanisms underlying neural tube closure defects caused by *Tet1* knockout and FA intake. For instance, the authors demonstrated alterations in metabolites involved in lipid metabolism, changes in DNA methylation of genes related to nervous system development, and reduced expression of genes involved in neurotransmission. However, since nervous system development and neurotransmission mainly occur post-neural tube closure, the relevance of these findings to neural tube closure remains unclear. The omics results should be discussed in direct relation to the neural tube closure phenotype.

Response: We agree that to relate our findings with neural tube closure, it would be more appropriate to perform all these metabolomic and epigenomic analysis in E8.5 headfold tissues. As we explained above in response to Reviewer 1, the initial focus of this study was to examine the phenotypes of *Tet1* KO mice in response to modified maternal FA status, which required us to examine the embryos at a post-neurulation stage. Based on the observation that NTD penetrance are not significantly altered by FA status, we decided to analyze the E11.5 embryonic brains collected to ask if neurodevelopmental pathways are perturbed. We hypothesize that the DNA methylation changes observed post-NTC might still overlap significantly with those occurring during NTC, given that methylation patterns can persist throughout development. Indeed, our findings at a post-NTC stage are suggestive of interactions that could happen earlier during NTC. In Discussion, we are cautious to highlight the caveats and limitations of our study. Whether or not the gene-environmental interactions we observed are relevant to neural tube closure *per se*, we believe we have unraveled an epigenetic mechanism regulating gene expression of solute carrier that could explain NTDs resistant to FA supplementation.

3. The involvement of *Slc46a1* in neural tube closure needs further investigation. It should be determined whether *Slc46a1* knockout causes neural tube closure defects and if overexpression of *Slc46a1* can rescue the *Tet1* knockout phenotype.

Response: Our RNA-seq analysis in E11.5 brains showed that while multiple transporters are down-regulated, *Slc46a1* is not among them. Instead, we observed promoter hypermethylation of *Slc46a1* at this developmental stage through amplicon-targeted bisulfite sequencing. Given the potential for functional redundancy among transporters, we are cautious about the likelihood that *Slc46a1* knockout or overexpression would directly impact neural tube closure or rescue the *Tet1* KO phenotype.

4. While pairwise comparisons were performed in Figures 4 and 5, a multiple comparison test among the nine groups is necessary to examine overlaps in Figures 4D and 5E.

Response: We consulted a biostatistician on how to perform more rigorous statistical analysis based on our research questions. Because the Venn diagram in Fig.4D only summarized the results of the DMR analysis of diet-specific comparisons between KO and WT in purely descriptive statements, we performed 2-way ANOVA with a post-hoc Tukey test to compare the methylation levels calculated for defined subsets of DMRs between groups using the genotype and diet as the two independent factors. For TET1-common DMRs (independent of diet; in the overlap of all three diet groups), *Tet1* genotype is an extremely significant factor accounting for methylation differences, whereas diet and the interaction between genotype and diet are not. On the other hand, for DMRs in the non-overlap and specific to each diet, the interaction between genotype and diet is significant in addition to genotype alone. The results of the 2-way ANOVA support our conclusions that TET1-common DMRs and TET1xFA diet-specific DMRs derived from distinct biological perturbations affecting different genomic loci, the former arising from loss of TET1 as the primary driver of DNA hypermethylation, the latter from an interaction between FA diet and *Tet1* genotype. To examine the overlapping DEGs in Fig.5F, we performed a one-sample binomial test to calculate the expected number of DEGs under independence of the dietary conditions and compared with the observed number. This analysis demonstrated that the observed proportion in the overlap exceeded the expected proportion under independence, suggesting that the observed overlap is significant and not due to random chance. Because only one test was performed, a multiple comparison test is not applicable.

5. The color scheme in Figures 1H and 3F is not distinguishable for colorblind individuals, such as this reviewer. They should use the same coloring scheme as in Figure 5C to improve accessibility.

Response: We thank Reviewer 2 for highlighting the accessibility issue with the color scheme used in Figures 1H and 3F. We have updated the color scheme in all heatmaps to a standard blue and red, consistent with Figure 5C.

Minor Points:

6. Figure 1A should include the Control genotype for comparison.

Response: We have included the NTD rates in WT and HET embryos in new Figure 1A.

7. All replicates should be shown in Figure 3E, rather than presenting only the average.

Response: The PCA plot of all individual replicates was originally shown in Supp.Fig.3C (now Appendix Figure S3C).

8. The terms "Tet1-specific" and "FA x TET1" in Figure 4D are misleading. It may be clearer to use "Tet1-dependent and FA-independent" and "both Tet1- and FA-dependent," respectively.

Response: We agree with Reviewer 2 that the terms in Fig.4D required further clarification. We have changed the terms to: "TET1-common" and "TET1xFA diet-specific". "TET1-common" refers to DMRs resulting from the loss of TET1, independently of FA status, whereas "TET1xFA diet-specific" refers to the collection of TET1-dependent DMRs modified specifically by each FA custom diet. We hope these changes make our classifications clearer to the reader.

9. On page 19, it is suggested that excess FA in the absence of TET1 may mimic FA deficiency. According to Figures 3B-D, the levels of folic acid, SAM/SAH, and Serine/Glycine differ between the 30 ppm and 0.1 ppm FA groups. Additionally, the CpG level in the promoter of *Slc46a1* (Figure 6E) exhibits different patterns between the 30 ppm and 0.1 ppm FA groups. The mechanisms by which differing FA availability leads to opposite downstream effects should be elucidated.

Response: We thank Reviewer 2 for pointing out the need to qualify our statement that excess FA in *Tet1* KO mimics FA deficiency. Our RNA-seq analysis suggests that in *Tet1* KO embryos, excess FA may paradoxically result in a compensatory response that resembles FA deficiency by downregulating solute carriers including possibly a folate transporter. Such a compensatory response may serve to protect the embryos from teratogenically high levels of maternal FA. Nonetheless, there are still significant differences in the metabolomic and methylome profiles between KO in 30 ppm and WT or HET in 0.1 ppm FA groups, suggesting that the downstream effects of excess and depleted FA in the context of *Tet1* loss are more complex. We have added this point in the revised Discussion, we have added the statements (page 27, line 630, font in blue).

10. According to Figure 4B, the normal diet (3 ppm FA) is the most effective condition for hyperDMRs, whereas Figure 2D suggests that FA deficiency has the most significant impact on the phenotype. Furthermore, in Figure 5A, the normal diet appears to have a limited effect on DEGs. The significance of hyperDMRs needs to be clearly explained in this context.

Response: We thank Reviewer 2 for pointing out these seemingly discrepant observations and the need to clarify the significance of the hyper DMRs. It is indeed a paradox why the normal 3 ppm FA diet is the most effective condition for observing hyper DMRs upon *Tet1* KO, instead of excess FA diet which is expected to drive more methylation reactions. In this revision, we have performed whole genome bisulfite sequencing (WGBS) of sample subsets from the 30 ppm FA and 3 ppm FA groups to resolve the issue of whether promoter-biased genome coverage by RRBS accounted for the apparent loss of TET1-dependent hyper DMRs when exposed to 30 ppm FA. Indeed, with full genome coverage, we observed equivalent numbers of hyper DMRs in the *Tet1* KO samples in both 30 ppm and 3 ppm FA conditions. GO analysis of the hyper DMRs in KOvsWT in 30 ppm and 3 ppm FA conditions shows similar pathway enrichments to those shown in our RRBS data, further validating our RRBS results. Specifically, the non-overlapping KO vs WT hyper DMRs in 30 ppm group are enriched for neurodevelopment terms. Interestingly, excess FA resulted in the re-distribution of TET1-dependent hyper DMRs from promoter proximal to more distal regions (new Appendix Figure S6), as we predicted. We discuss that this observation can also be consistent with a state of pseudo-FA deficiency in *Tet1* KO exposed to excess FA, in which down-regulation of FA intake may divert a more limiting pool of one-carbon units towards methylation of CpG shore and distal regions in the absence of TET1. Nonetheless, the loss of *Tet1* is the primary driver of hyper DMRs under all dietary FA conditions, as shown by the PCA and hierarchical cluster plots in Appendix Figure S4B-C. FA deficiency would also promote DNA hypomethylation, in agreement with its most significant impact on the phenotype when coupled with *Tet1* haploinsufficiency

(Figure 2C). The limited impact of normal diet on DEGs is likely due to the already low expression levels of *Tet1* at E11.5 stage. As discussed above, the poor association between DNA methylation and gene expression readouts at E11.5 is likely because the methylation changes reflect earlier epigenetic events that persist and propagate through development, whereas transcriptional status are much more temporal and dynamic. Thus, if FA diet x *Tet1* genotype interactions affect NTC, we may still capture DNA methylation changes at a post-NTC stage but not always the gene expression changes. Nonetheless, the detection of DEGs that remained down-regulated in E11.5 *Tet1* KO embryos under 30 ppm FA condition is significant.

Referee #3:

1. Does this manuscript report a single key finding? YES. Loss of TET1 function leads to blunted molecular response to maternal FA supplementation, thus *Tet1*^{-/-} embryos resemble NTD cases that are non-responsive to FA-supplementation.

2. Is the reported work of significance (YES), or does it describe a confirmatory finding or one that has already been documented using other methods or in other organisms etc (NO)? YES

3. Is it of general interest to the molecular biology community? YES, this study employs a lot of controls and extensive pair-wise comparison in order to distinguish phenotypes (molecular and developmental phenotypes) that are caused by 1) environmental exposure, 2) genotypes, and 3) interactions between exposure and genotypes.

4. Is the single major finding robustly documented using independent lines of experimental evidence (YES), or is it really just a preliminary report requiring significant further data to become convincing, and thus more suited to a longer-format article (NO)? YES

Response: We thank Reviewer 3 for these highly positive assessments.

In this manuscript, Chen et al. sought to understand how maternal folic acid (FA) levels influence embryonic brain development, specifically neural tube closing, in the context of TET1 loss of function. This study is partly motivated by high numbers of human neural tube defects (NTDs) that are resistant to FA supplement. As folate feeds into the one-carbon metabolism pathway that produces the methyl donor S-adenosylmethionine, the authors posit that interactions between genetic and the maternal FA environment are facilitated through the 5mC dioxygenase TET1. TET1 loss of function mouse models show varying degrees of NTD penetrance that is dependent on the homogeneity of the genetic background of the mouse line. The authors spent considerable effort to characterize the baseline NTD rates in the highly penetrant CD1-*Tet1*^{-/-} line and the less-sensitive, but more congenic (thus, more appropriate for downstream molecular analyses) 129S6-*Tet1*^{-/-} line. Overall, the authors conclude that the loss of TET1 renders embryos less responsive to molecular changes elicited by FA supplementation.

This may be explained by the blunted delivery of excess FA to 129S6-Tet1^{-/-} embryos due to decreased expression of Slc46a1. Interestingly, FA-deficiency does not seem to exacerbate the NTD or molecular phenotypes of 129S6-Tet1^{-/-} embryos. I have a couple major comments that need to be addressed and several minor comments that will help to streamline the various models used in this study which I have included below:

Major comments:

1. Beginning in Figure 2, there are significant discussions on the effect of Tet1 haploinsufficiency on the responsiveness to maternal FA levels, including a stark subset of HET-specific DEGs in response to FA-deficiency. In this manuscript or the preceding Khoueiry et al., 2017 where the Tet1^{tm1Koh} was initially generated, I cannot find mRNA or protein levels of WT vs HET vs KO to support that 1) no full-length protein is generated in the KO or 2) TET1 is actually present at reduced level in 129S6-Tet1^{+/-} embryos or embryonic brains. In fact, as the authors pointed out in Supplementary Figure 5B, this gene-trap allele still allows for expression of the short somatic Tet1 isoform from a downstream TSS in the KO embryonic brains. A careful validation of the gene product of the Tet1^{tm1Koh} allele should be included, including a Western blot using both C- and N-terminus TET1 antibody, preferably on embryonic brains of relevant developmental stage. In relation to this point, have the authors checked expression levels of Tet1 (from both the canonical TSS and the downstream alternative TSS) in Tet1^{-/-} as the line was outcrossed to CD1 and as the line was backcrossed to the 129S6 background (in generations where NTD was high in penetrance N5-9 and when the 129S6 was considered congenic). Is there a correlation between expression of the short isoform of Tet1 (if the full length Tet1 is indeed absent) and the percentage of KO-embryos with NTD?

Response: We thank Reviewer 3 for raising this important point. The results of Khoueiry et al. were based largely on an earlier *Tet1* gene trap mouse strain, which was subsequently validated by a new gene targeted *Tet1*^{tm1Koh} knockin-knockout strain. The validation of complete KO of *Tet1* full length transcripts in ESC, iPSCs and NPCs by the *Tet1*^{tm1Koh} allele based on RNA-seq and Western blots can be found in our subsequent publications (Suppl.Fig.7 in Bartocetti et al, 2020; Suppl.Fig.1 in van der Veer et al. 2023). In this revision, we had added 1) qPCR analysis of ESCs, NPCs and E11.5 brains to evaluate both the full length and short isoform of *Tet1*. Our results confirm that the full length *Tet1* transcript is indeed absent in *Tet1* KO brain at E11.5, while the short isoform is present but at low levels in both WT and *Tet1* KO E11.5 brains (new Appendix Figure S7C). We included *Tet1* KO embryos that are either normal or affected by brain malformation or NTDs in the analysis but did not observe a correlation between the expression of *Tet1* short isoform and the phenotypes. Thus, the TET1 short isoform is activated at low levels by E11.5 but is not sufficient to rescue DNA hypermethylation caused by the loss of full-length protein. 2) We performed western blot on WT and KO ESCs and E11.5 brains with a TET1(N-terminal) antibody. We observed an absence of TET1 protein in both WT and KO E11.5 brains, suggesting TET1 protein level, including that of the short isoform, is undetectable at this stage (Appendix Figure S7D). Unfortunately, the TET1(C-terminal) antibody (Millipore) used in our previous publications is no longer commercially available. We conclude that TET1 expression is much lower in the E11.5 mouse embryonic brain than in ESCs and E6.5 epiblast, both at RNA and protein levels. Therefore, the neural tube closure defects and subsequent DNA methylation changes detected at E11.5 are primarily the result

of TET1 loss-of-function in the primed epiblast. Low level expression of a somatic short isoform by E11.5 is not sufficient to reverse any of these phenotypes.

2. The RRBS was conducted using oxidative bisulfite (ox-BS) pipeline. This should allow for delineation of 5mC vs 5hmC within the library through subtractive analysis methods if the BS-libraries were sequenced in parallel. I will preface this comment by mentioning that I recognize the low-level of 5hmC in the embryonic brain and that most 5hmC accumulation occurs postnatally in post-meiotic neurons. In this study, the dietary (excess or deficient FA) effects on the methylome, and to some degree gene expression, is only observed in the WT and HET, and not in the KO (Figure 4C, 5D right panel). This suggests that TET1's activity (5hmC generation) facilitates response to differing levels of maternal FA. Additionally, since there is very little relationship between DNA methylome changes to the observed changes in gene expression, 5hmC profiling should be conducted to obtain a comprehensive view on the interactions between 1-C metabolism, Tet1 genetic status, and NTD phenotype. While the literature is scant, there are emerging findings that NTDs are associated with global decreased in global 5hmC levels (PMIDs: 36199572; 31179819).

Response: We thank Reviewer 3 for this insightful comment on 5hmC, which we have not profiled in this study. We have previously shown that 5hmC is barely detectable in the E8.5 mouse embryo (Khoueiry et al. 2017), suggesting that the presence of 5hmC may be highly locus-specific or not required during the initiation of neural tube closure. Due to budget constraints, we performed oxidative RRBS without conventional BS (non-oxidative) reactions in parallel, so we cannot directly assess 5hmC from the current data. In this revision, we performed a 5hmC dot blot on ESCs and E11.5 brain from both WT and KO samples. Our result shows that 5hmC content in WT E11.5 brains is low and near background levels, compared to much higher levels in WT ESCs (Appendix Figure S7E). Nonetheless, we agree that 5hmC may have an interesting role in mediating the interaction between *Tet1* and FA status in WT and HET embryos, but this will have to be assessed earlier in the E6.5 epiblast.

Minor points:

1. As a reader, I consider CD1-Tet^{-/-} vs 129S6-Tet1^{-/-} to be completely different mouse lines when I interpret the result. While in many figures the genetic backgrounds are included in the labeling, they are missing in several of them and only included in the figure legends. Please consider always including the genetic background in labeling the figures (e.g. Figure 1C, 1H, Figure 3, 4, 5, and 6).

Response: We thank Reviewer 3 for pointing out the missing labels of genetic backgrounds in some figures. We have updated the figures with strain information (Figure 1C, 1D, 1G-I; Figure 3, 4, 5, and 6E).

2. Have the authors measured FA metabolites/FA levels in serum of the dam?

Response: We have not measured serum FA levels of the dam. We consulted with our collaborators on the feasibility of this measurement and advised that technical limitations preclude accurate measurement of FA levels in the serum.

3. Line 83 should read: "...DNA demethylation via the iterative oxidation..."

Response: We have updated this sentence in the manuscript to: "...DNA demethylation via the iterative oxidation...".

4. Line 128, should E10.5 be E9.5 instead as that is the time point analyzed in Figure 1C.

Response: We thank Reviewer 3 for pointing out the discrepancy between the text and Fig.1C. To address this, we have included an additional image in new Appendix Figure S1A, which shows the complete closure of posterior neuropore at E10.5. This image clarifies the closure time-point and aligns the figure with the textual description.

5. Please refer to the strain of KO in line 244.

Response: We have updated the sentence to: "Like in CD1 mice, open NTDs in 129S6.Cg *Tet1*^{-/-} (KO) embryos...."

6. Figure 3B should be ordered the same way as Figure 3C and 3D.

Response: We agree and have updated the sample group order in Fig.3B (new Fig. 3C) to be like those in Figure 3C and 3D (new Fig. 3D and 3E).

7. Figure 3F: I think the ordering of this heatmap should be changed in order to communicate the result better. I think the authors are trying to relay that the FA-dose effect is seen in WT and HET but not in KO. In order to immediately see this dose-dependent effect on metabolites the top layer of ordering should be genotype, then intercalate the dose as the second layer.

Response: We appreciate the suggestion for improving the display of the heatmap in Figure 3F (new Fig. 3G). We experimented with reordering the heatmap by genotype as the top layer and FA dose as the second layer. However, we found that the original ordering highlights the main findings more effectively: the effect of FA levels on metabolites is dominant, while the impact varies in more subtle ways across different *Tet1* genotypes. The original order allows us to show more clearly the loss of metabolites in *Tet1* KO specifically in the 30 ppm FA diet group.

8. Related to Figure 4A, can the authors provide violin plots of global methylation levels to complement these metagene plots so that readers can quickly assess changes in global methylation levels between genotypes and diets.

Response: We have updated new Fig. 4A to display methylation levels stratified by genotypes and diets, highlighting hypermethylation in *Tet1* KO and the effect of *Tet1* haploinsufficiency. Additionally, we have included cumulative distribution plots of methylation levels of all 1 kb tiles across the genome (new Fig. 4B). We have added the violin plots of global methylation levels for various genomic regions (1 kb tiles, CpGi, 1-5 kb upstream and downstream, promoters, exons and introns) in new Appendix Figure S4D. We hope these plots provide finer details of differences depicted in the metaplots.

9. Related to Figure 4B, are there overlap between the DMRs found in KO vs WT (3ppm FA) and those found in HET vs WT (3ppm FA)? Where in the genome are these overlapping DMRs located in?

Response: We performed a Venn-diagram analysis of the hyper DMRs identified in KO/HETvsWT (3ppm FA) comparisons and found a significant overlap of 3759 regions (new Appendix Figure S5A). These overlapping DMRs are predominantly located in CpG islands and associated with promoter regions (new Appendix Figure S5B). Additionally, GO analysis of these overlapping regions revealed enrichment in terms related to the regulation of neurogenesis, axonogenesis and mesenchymal development (new Appendix Figure S5C), similar to TET1-common hyper DMRs.

10. Related to Figure 4C, there are many DMRs that are found in WT in response to excess or deficient maternal FAs. What are the GO terms for these exclusively FA-driven DMRs and are there specific genomic compartments that these FA-driven DMRs located in?

Response: In the original manuscript, GO terms for the DMRs that are found in WT in response to excess or deficient maternal FAs were originally shown in Suppl. Figures 4D and 4E respectively. We moved these figures to the main Figures 4H and 4I, because we think that the impact of excess or deficient maternal FA levels on WT embryos are of interest. The GO analysis of hyper DMRs found in WT in respond to FA excess showed enrichment for neurodevelopment terms and were highly enriched in promoter regions (Figure 4H, Appendix Figure S5F). Both hyper and hypo DMRs induced by depletion were associated with GO terms for development and Wnt signaling, the latter being the top GO term enriched among hypo DMRs (Fig. 4I). Furthermore, the hyper DMRs induced by FA deficiency are more enriched for distal loci, while the hypo DMRs are more enriched in CpG islands and CpG shores of promoters (new Appendix Figure S5G). These findings would be consistent with the expectation that DNA hypomethylation induced by FA deficiency would have a more direct impact on developmental gene expression.

11. I do not agree with the rationale for DEGs overlapping in Figure 5E. I do not think this is informative to identify "genes that are responding to dietary FA fluctuations". Instead of Figure 5E and 5F, can the authors instead provide deepTools analyses of where the DMRs are located with respect to DEGs (distance from DEGs, or metaplots of where in DEGs). Or alternatively, where TET1 is found (from the TET1 EpiLC ChIPseq for example) with respect to DEGs.

Response: We thank Reviewer 3 again for this helpful suggestion. We have added a heatmap showing the promoter methylation levels with respect to all DEGs in new Fig.5D. Indeed, this visualization more clearly illustrates the relationship between methylation and gene expression. We observe a correlation between hypermethylation and the downregulation of gene expression in *Tet1* KO, independent of sex and phenotype. We agree that Figure 5F and subsequent figures are not informative in identifying genes responding to dietary FA fluctuations, but these Venn diagram analyses of the DEGs support our hypothesis that the excess FA in *Tet1* KO mimics the effect of FA deficiency on target genes.

12. For ESC to antNPC differentiation, the authors mention in line 450 of using "defined culture media containing high FA supplements (>5mg/L)". Is this concentration of FA supplement a requirement for antNPC differentiation or is it a culture condition that is supposed to mimic high maternal FA environment?

Response: The supplement of FA in the culture medium is standard practice and essential for cell maintenance and proliferation. The high concentration of FA used during ESC to NPC differentiation is not specifically required for differentiation but reflects typical culture conditions that ensure optimal cell growth and viability.

Dear Kian,

Thank you for submitting your revised manuscript. It has now been seen by all of the original referees.

My apologies for the delay in getting back to you - it took longer than anticipated to receive the referee reports.

As you can see, the referees find that the study is significantly improved during revision and recommend publication. While we appreciate the comments of referee #2, we also find that it is not necessary to restructure the manuscript into two distinct ones for publication here. However, I need you to address the points below before I can accept the manuscript.

- Please address the minor concern of referee #3.
- Please provide 3-5 keywords for your study. These will be visible in the html version of the paper and on PubMed and will help increase the discoverability of your work.
- Please make the datasets GSE275393 and GSE275401 publicly available and remove the review tokens from the manuscript texts. Also, URLs that directly resolve to the datasets need to be provided in the Data Availability section (not just to the homepage of the database).
- As per our guidelines, Data Availability section is reserved for the new primary dataset that is generated in this study and deposited in a public data repository. We note that datasets GSE71744 (Khoueiry et al. 2017) and GSE214845 (van der Veer et al. 2023) are existing publicly available datasets. Therefore, please remove the information regarding GSE71744 (Khoueiry et al. 2017) and GSE214845 (van der Veer et al. 2023) from the Data Availability section. Instead, please cite GSE71744 (Khoueiry et al. 2017) and GSE214845 (van der Veer et al. 2023) in the text in the form of data citation -Also, please cite the studies including these datasets separately as well in line with the example I give below.

In text: (Skruber et al, 2020; Data ref: Skruber et al, 2020).

In the reference list:

Dataset: Skruber K, Warp PV, Shklyarov R, Thomas JD, Swanson MS, Henty-Ridilla JL, Read TA, Vitriol EA (2020) Gene Expression Omnibus GSE149870 (<https://www.ncbi.nlm.nih.gov/geo/query/acc.cgi>). [DATASET]

Paper: Skruber K, Warp PV, Shklyarov R, Thomas JD, Swanson MS, Henty-Ridilla JL, Read TA, Vitriol EA (2020) Arp2/3 and Mena/VASP Require Profilin 1 for Actin Network Assembly at the Leading Edge. *Curr Biol* 20;30(14):2651-2664.e5.

Please see <https://www.embopress.org/page/journal/14693178/authorguide#referencesformat>

- Please rename the 'Competing interests ' as 'Disclosure Statement and Competing Interests'.
- Please rename the 'Materials and Methods' as 'Methods'.
- Please move Tables 1-3 after the main figure legends.
- Please remove the 'Author contributions' section from the manuscript.
- We note the phrase 'data not shown' on pages 23 and 24, which is not allowed as per journal policy. Please either show the data or remove the statements.
- Please include the funding information in the Acknowledgements section. As per journal policy, the funding information needs to be complete in both our manuscript tracking system and the manuscript file. Currently, we note the following:

Missing in the manuscript tracking system:

- FWO grant numbers G092518N, G0C6820N
- KU Leuven Internal Funds C14/21/117
- U.S. National Institute of Environmental Health Sciences Gulf Coast Center for Precision Environmental Health pilot award P30ES030285
- PhD fellowship 11E7920N
- KU Leuven postdoctoral mandate PDMT2/23/083
- CSC grant number 202004910440
- MSCA SoE FWO postdoctoral fellowship 673343/12ZZE23N
- U.S. National Institute of Health R01 grant HD100535

Missing in the manuscript file:

- EC | Horizon Europe | Excellent Science | HORIZON EUROPE Marie Skłodowska-Curie Actions (MSCA)
- The figures need to be resubmitted as individual production quality figure files as .eps, .tif, .jpg (one file per figure).
- Expanded View Tables S1-S3 are provided in a PDF. Each table should be uploaded as a separate file as Table EV1-EV3 and the callouts in the manuscript need to be updated. The legends for these tables are missing and each should be provided in its corresponding file.
- Please remove the Reagents and Tools table from the manuscript and upload it as a separate (word) document.
- Please resubmit source data as one file per figure.
- Our production/data editors have asked you to clarify several points in the figure legends:
 - Please note that the exact p values are not provided in the legends of figures 1d; 3d-e; 5f.
 - Please indicate the statistical test used for data analysis in the legends of figures 4f, h-i; 5a-b, e.

- Please note that the box plot needs to be defined in terms of minima, maxima, centre, bounds of box and whiskers, and percentile in the legend of figure 1d.
- Please note that information related to n is missing in the legends of figures 5h-i.
- Please note that the error bars are not defined in the legends of figures 1h; 5i.
- Papers published in EMBO Reports include a 'synopsis' and 'bullet points' to further enhance discoverability. Both are displayed on the html version of the paper and are freely accessible to all readers. The synopsis includes a short standfirst summarizing the study in 1 or 2 sentences (max 35 words) that summarize the paper and are provided by the authors and streamlined by the handling editor. I would therefore ask you to include your synopsis blurb and 3-5 bullet points listing the key experimental findings.
- In addition, please provide an image for the synopsis. This image should provide a rapid overview of the question addressed in the study but still needs to be kept fairly modest since the image size cannot exceed 550 (width) x 300-600 (height) pixels.

Thank you again for giving us to consider your manuscript for EMBO Reports, I look forward to your minor revision.

Kind regards,

Deniz

--

Deniz Senyilmaz Tiebe, PhD
Senior Scientific Editor
EMBO Reports

Referee #1:

Most points of the original critique have been addressed and the manuscript by Chen et al. has been sufficiently revised to warrant publication.

Referee #2:

The authors have not adequately addressed the primary concern raised by this reviewer in previous comments: namely, 'it remains unclear how the findings from these analyses contribute to an understanding of neural tube closure defects.' Based on the responses to comments 2 and 3, it appears that the experiments in Figures 3, 4, 5, and 6 are focused on investigating the effects of Tet1 and folic acid (FA) on the embryos and their cells, rather than directly elucidating the mechanisms underlying neural tube defects (NTDs). This, in turn, may contribute to what reviewer #1 has pointed out as 'a separate problem with the study: the lack of cohesion.'

In light of this, this reviewer suggests that the manuscript be restructured into two distinct papers: one focusing on the NTD phenotype, as presented in Figures 1 and 2, and the other focusing on the omics analysis, as presented in Figures 3, 4, 5, and 6. This approach may enhance the clarity and overall cohesion of the paper.

Referee #3:

The authors have sufficiently addressed my original concerns. The authors added figures or supplemental figures as requested and rationalized limitations when the requested experiments were not able to be conducted.

- I think there is a mistake in the labeling of legends for Figure 2B and C in which the color codes for NTDs and Normal are switched around.

All editorial and formatting issues were resolved by the authors.

Dr. Kian Peng Koh
Stem Cell Institute Leuven
Department of Development and Regeneration
Belgium

Dear Kian,

Thank you for submitting your revised manuscript. I have now looked at everything and all is fine. Therefore, I am very pleased to accept your manuscript for publication in EMBO Reports.

Congratulations on a nice work!

Kind regards,

Deniz

--
Deniz Senyilmaz Tiebe, PhD
Senior Scientific Editor
EMBO Reports
